# Memorization in Graph Neural Networks

**Adarsh Jamadandi**[1][*] **Jing Xu**[2]**, Adam Dziedzic**[2]**, Franziska Boenisch**[2]
CNRS IRISA[1], CISPA Helmholtz Center for Information Security[2]
adarsh.jamadandi@irisa.fr, {jing.xu, adam.dziedzic, boenisch}@cispa.de

## Abstract

Deep neural networks (DNNs) have been shown to memorize their training data, but similar analyses for graph neural networks (GNNs) remain under-explored. We introduce `NCMemo` (`Node Classification Memorization`), the first framework to quantify label memorization in semi-supervised node classification. We establish an inverse relationship between memorization and graph homophily, *i.e.,* the tendency of connected nodes to share labels or features. Lower homophily significantly increases memorization, indicating that GNNs rely on label memorization when learning less homophilic graphs. We then analyze GNN training dynamics and find that increased memorization in low-homophily graphs is tightly coupled to GNNs' implicit bias toward using graph structure. When structure is less informative, models instead memorize node labels to minimize training loss. Finally, we show that nodes with higher label inconsistency in their feature-space neighborhood are more prone to memorization. Based on these insights, we investigate graph rewiring as a mitigation strategy. Our results show that rewiring reduces memorization without harming model performance, while also lowering the privacy risk for previously memorized data points. Thus, our work advances understanding of GNN learning and supports more privacy-preserving GNN deployment.

## 1 Introduction

Graph Neural Networks (GNNs) are a class of deep learning algorithms that adopt the paradigm of message passing [27, 55, 24, 14] to learn on graph-structured data. GNNs have found successful applications in diverse domains such as chemistry [52], biology [12], high-energy physics [58], and even contribute to the discovery of novel materials [42]. Despite their empirical success, the inherent limitations of GNNs continue to be an active area of research, one of which is the challenge of generalization. In deep neural networks (DNNs), memorization, the model's ability to *remember* individual training data points, and how it benefits generalization has been explored [21]. While recent studies [60, 64, 3, 39] have focused on establishing generalization bounds and errors for GNNs, investigation of GNN memorization is notably scarce. A primary reason for the gap in the understanding of memorization for GNNs is the inherent difficulty in conceptualizing and defining per-sample memorization. In other domains, such as computer vision, samples are independent from each other, which allows for a straightforward definition of per-sample memorization. However, in a node classification setting, the GNN operates on a *single graph* where all nodes are inter-connected. This makes analyzing the model's ability to memorize individual nodes challenging, as it requires modeling a complex interplay of factors: the node's own features, the features of its neighbors, the information content of its label relative to its neighborhood [49], the overall graph homophily level, and the GNN's training dynamics.

In light of these challenges, we propose a systematic framework for measuring memorization in node classification. Our approach, called `NCMemo` (`Node Classification Memorization`),

---

[*]work carried out at CISPA.

39th Conference on Neural Information Processing Systems (NeurIPS 2025).

tackles these challenges directly by characterizing memorization from three perspectives: (i) at the node level, by taking into account the feature and label information; (ii) at the graph level, by studying memorization with respect to graph homophily; and finally (iii) the training dynamics, by understanding how memorization affects the ability of the GNNs to learn generalizable functions. Our proposal is inspired by the *leave-one-out* [21, 63] definition of memorization which involves comparing the predictive behavior of models trained with and without specific data samples. A significant change in this behavior, particularly an increased confidence when the sample is included in training, signals that the sample has been memorized. This *behavioral gap* helps identifies nodes potentially prone to memorization.

At first glance, extending this idea to graphs may seem challenging, since nodes are interdependent and masking a single label could lead to information leakage. However, this setup aligns with the standard semi-supervised node classification paradigm, where models access all node features and the full graph structure but only a subset of labels during training. Our leave-one-out procedure is fully consistent with this setting, *i.e.,* we mask one node's label, keep all other information fixed, train models with and without that label, and measure the resulting change in predictive behavior. Further discussion is provided in Appendix A. Our study reveals several key insights: we find a novel inverse relationship between memorization rate (the number of nodes that get memorized) and graph homophily, *i.e.,* the nodes in lower homophily graphs exhibit higher memorization. To explain the emergence of this memorization behavior, we analyze the training dynamics of GNNs in the over-parametrized regime through the lens of Neural Tangent Kernel [31, 4, 66]. We find that the tendency of GNNs to be excessively reliant on the graph structure, even when it is uninformative, as in the case with lower homophilic graphs, results in a model memorizing node labels. We also investigate memorization at the node level by proposing a novel metric, *i.e.,* Label Disagreement Score (LDS), which helps identify why certain nodes are susceptible to memorization. Overall, our analysis suggests that nodes with higher label inconsistency in their feature space, *i.e.,* atypical samples, experience higher memorization. Finally, we show that memorized nodes are significantly more vulnerable to privacy leakage [59] than non-memorized ones. Motivated by the observed link between graph homophily and memorization, we explore graph rewiring [61] as a mitigation strategy. Specifically, following [54], we rewire edges based on the cosine similarity between connected nodes. This reduces memorization and, as we show, the related privacy risks for memorized nodes, while preserving model performance.

Overall, we perform extensive experiments on various real-world datasets and semi-synthetic datasets [68] with various GNN backbones such as SGC [65], GCN [37], GraphSAGE [28], GATv2 [13] and Graph Transformer (GT) to confirm our findings. In summary, we make the following contributions.

1. We introduce `NCMemo`, a label memorization framework inspired by the *leave-one-out* memorization style [21] for the challenging context of semi-supervised node classification.

2. We reveal an interesting inverse relationship between the memorization rate and the graph homophily level: as the graph homophily increases (and consequently label informativeness [49] increases), the memorization rate decreases significantly.

3. We explain memorization's emergence by analyzing GNN training dynamics. We find that GNNs' tendency to utilize graph structure, even when it's unhelpful in low-homophily scenarios, directly leads to memorization as they strive for low training error.

4. We propose the Label Disagreement Score (LDS) to characterize memorized nodes. This metric, measuring local label-feature inconsistency, shows that nodes with higher LDS are significantly more prone to memorization.

5. Finally, we demonstrate the privacy risks arising from memorization in GNNs and show that graph rewiring as a promising initial strategy for mitigating this risk.

## 2   Background and Related Work

**Limitations of Graph Neural Networks.** Although GNNs are being used in various domains, our understanding of the mechanisms governing how GNNs learn and generalize is still in its infancy. More recently, attention has shifted towards understanding GNN generalization. Approaches in this area include studying generalization gaps [60, 64], leveraging mean-field theory to study generalization errors [3], establishing convergence guarantees [36] and studying the interplay of generalization and

model expressivity [39]. Concurrently, other works analyze GNN training dynamics. For instance, Mustafa et al. [44] show how standard weight initialization schemes such as Xavier [26] might lead to poor gradient flow dynamics and consequently, diminished generalization performance in Graph Attention Networks (GATs) [62, 13]. To address this, they propose an improved initialization scheme for GATv2 [13] that enhances gradient flow and trainability. Furthermore, a recent work [66] shows the utility of Neural Tangent Kernel (NTK) [31, 4] to study GNNs in function space, characterizing how optimization introduces an implicit graph structure bias which is encoded in the notion of *Kernel-Graph Alignment* that aids learning generalizable functions.

**Memorization in Deep Neural Networks.** The phenomenon of memorization in deep learning—where models learn to remember specific training examples—has been extensively studied in the context of deep neural networks (DNNs) [67, 5, 21, 6, 40, 11, 63, 8, 38]. It has been shown that DNNs can memorize even random labels [67]. Different memorization metrics have been proposed for supervised [21] and self-supervised learning models[63], showing that memorization in DNNs is required for generalization. Memorization has also been associated with multiple risks to the privacy of model's training data [17, 16]. However, similar studies on data memorization in GNNs are lacking. In this work, we aim to fill this gap.

**Overfitting vs Memorization in GNNs.** A work that thematically aligns with the goals of this paper is [9] which studies how the GNNs' inductive bias towards graph structure can be problematic. The work demonstrates that GNNs may overfit to the graph structure even when ignoring it would yield better solutions, leading to poor generalization, particularly in graph classification. Our work on the other hand, focuses on the related but distinct phenomenon of memorization. Unlike overfitting where the model shows poor generalization beyond the training set, memorization involves using spurious signals to *remember* specific examples [38], which can be beneficial for generalization [21].

## 3 Our `NCMemo` **Label Memorization Framework**

In this section, we present our framework `NCMemo`, which is inspired by the *leave-one-out* definition of memorization proposed by Feldman [21]. Let $f : \mathbb{R}^n \to \mathbb{R}^d$ be a GNN model trained on the graph $G = (\mathbf{X}, \mathbf{A})$, where $\mathbf{X}$ is the node feature matrix and $\mathbf{A}$ is the adjacency matrix, using a GNN training algorithm $\mathcal{T}$ (e.g., GCN [37]). Let $\mathcal{S} = \{v_i, y_i\}_{i=1}^m$ be a labeled training dataset , where $v_i$ is a node and $y_i$ its true label. Let $f(v_i)$ denotes a predicted probability of model $f$ for the node $v_i$. And let $v_i$ be a target node from dataset $S$. Finally, let GNN models $f \in \mathcal{F}$, $g \in \mathcal{G}$ be trained with a GNN algorithm $\mathcal{T}$ on $S$ and $S \setminus v$ (dataset $S$ with $v$ removed), respectively. We calculate the memorization score $\mathcal{M}$ for a target node $v_i$ as:

$$\mathcal{M}(v_i) = \mathop{\mathbb{E}}_{f \sim \mathcal{T}(S)}[\Pr[f(v_i) = y_i]] - \mathop{\mathbb{E}}_{g \sim \mathcal{T}(S \setminus v_i)}[\Pr[g(v_i) = y_i]]. \tag{1}$$

Our objective is to measure the memorization for candidate nodes ($v_i$). Since model $f$ has seen the candidate nodes as part of its training corpus, it is expected that $f$ shows higher confidence in these nodes than model $g$. We define the **memorization rate** (`MR`) as

$$\mathtt{MR}(\%) = \frac{1}{|\mathcal{S}|} \sum_{v_i \in S} \mathbb{I}(\mathcal{M}(v_i) > \tau) \times 100, \tag{2}$$

where $\mathbb{I}(\cdot)$ is the indicator function and $\tau = 0.5$ as a predefined threshold, chosen according to the definition proposed in [21].

Since a full leave-one-out evaluation is computationally prohibitive, as it requires retraining for every node. To make this tractable, we follow the efficient setup detailed in [63]. We partition the training data into three disjoint subsets: $S_S$, $S_C$ and $S_I$. We use these sets to train two models. Model $f$ is our target model whose memorization we want to evaluate, while $g$ is an independent model that we use to calibrate according to Equation (1), following the leave-one-out-style definition of memorization by Feldman [21]. Model $f$ is trained on $S_S \cup S_C$, and model $g$ is trained on $S_S \cup S_I$, where $S_C$ is the candidate set, *i.e.,* the nodes whose memorization we want to quantify. By the setup of the experiment, models $f$ and $g$ are both trained on $S_S$, but $S_C$ is only used to train $f$. Therefore, the difference in behavior between model $f$ and $g$ on $S_C$ results from the fact that $f$ has seen the data and $g$ has not, allowing us to quantify the memorization. Including $S_I$ into the training set of $g$ makes $f$ and $g$ have the same number of training data points, which allows model $g$ to reach the performance of model $f$. This makes sure that differences in behavior on $S_C$ are not due to model performance

differences. Applied to our node classification setting, we include node $v_i$ for training model $f$ and for $g$ we do not. We can measure the memorization on this node as the expectation of behavior of models $f$ and $g$ on predicting the label of $v_i$ correctly as $y_i$. Intuitively, if $f$ is, on expectation, more capable of correctly predicting the label than $g$, this must result from the sole difference between the models, namely that $f$ has seen $v_i$ and $g$ has not. It hence quantifies $f$'s memorization on $v_i$. In the next section, we explain the mechanisms that drive GNN models to memorize and also provide insights into emergence of memorization behavior.

## 4  Mechanisms of Memorization : Graph Structure to Training Dynamics

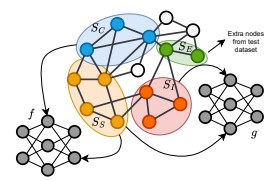

In this section, we study the various factors that lead to memorization in GNNs. Particularly, we identify three factors, namely the graph homophily, the training dynamics and the label disagreement at the node level as the key contributors to memorization. To evaluate our label memorization framework, we follow the setup in [63] to make the evaluation computationally tractable by approximating the memorization metric over multiple samples at once. Specifically, we divide the training set into three disjoint partitions—shared nodes ($S_S$), candidate nodes ($S_C$) and independent nodes ($S_I$). Subsequently, we train model $f$ on $S_S \cup S_C$ and model $g$ on $S_S \cup S_I$. Figure 1 illustrates the data partitioning in our label memorization framework.

Figure 1: **Data partitioning in our label memorization framework.**

We begin our evaluations by using a synthetic dataset benchmark, where we can control the specific properties of the graphs, *e.g.,* homophily level. The synthetic benchmark is the `syn-cora` dataset proposed in [68], which contains graphs with different homophily levels, the node features for the graphs are assigned from a real-world graph such as Cora [41]. We choose five graphs with increasing homophily levels $h = \{0.0, 0.3, 0.5, 0.7, 1.0\}$, where, for example `syn-cora-h0.0` indicates a synthetic graph with homophily level $0.0$. Note that, low homophilous graphs are also called as *heterophilic* graphs, where the connected nodes have different labels/features.

### 4.1  Highly Homophilous Graphs Exhibit Lower Memorization Rates

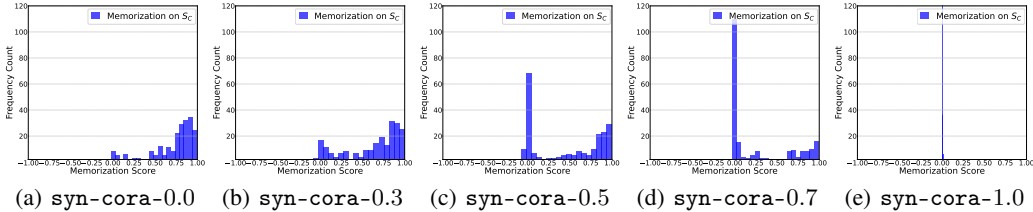

(a) `syn-cora-0.0`  (b) `syn-cora-0.3`  (c) `syn-cora-0.5`  (d) `syn-cora-0.7`  (e) `syn-cora-1.0`

Figure 2: **Memorization Scores for $S_C$ on `syn-cora`:** Comparison of memorization score across `syn-cora` graphs with increasing homophily levels trained with GCN.

We train a three-layered GCN [37] on each graph of `syn-cora` and calculate the memorization scores based on Equation (1). We plot the memorization scores for the candidate nodes ($S_C$) across different graphs in Figure 2. We can see that the memorization score and memorization rate is higher for heterophilic graphs than homophilic graphs.[2] This result unveils an interesting connection between the memorization rate, see Equation (2) and the graph homophily level ($h$), leading us to our first novel insight. We formalize this in a proposition below.

**Proposition 1 (Inverse relation between memorization rate and homophily.)** *The rate of memorization is inversely proportional to the graph homophily level ($h$). As the homophily level increases ($\uparrow$), the memorization rate tends to zero ($\downarrow$).*

To investigate the claim made in this proposition, we plot the graph homophily level, the node label informativeness (NLI) which measures the amount of information the neighboring nodes give to

---

[2]In our generated `syn-cora` dataset, the number of nodes is the same for all generated graphs so it is straightforward to compare the memorization rate between graphs of different homophily levels.

predict a particular node's label [49], and the memorization rate for the candidate nodes in Figure 3(a). The results highlight that as the graph becomes highly homophilic ($h = 1.0$), the memorization rate drops to 0. To see the relationship more clearly, in Figure 3(b) we plot only the homophily vs memorization rate with a red line highlighting the trend. This confirms our hypothesis that one of the mechanisms dictating the memorization behavior in GNNs is the graph homophily level.

## 4.2 GNNs' Structural Bias and Kernel Alignment Dynamics Explains the Emergence of Memorization

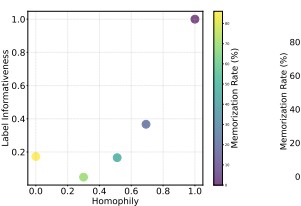

(a) Homophily vs Label Informativeness vs Memorization Rate.

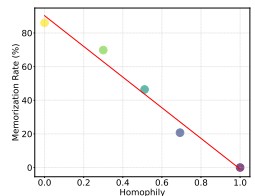

(b) Homophily vs Memorization Rate.

Figure 3: **Inverse relationship between Memorization Rate and Homophily.** We show results for GCN trained on the `syn-cora` dataset. As the graph homophily increases, the memorization rate decreases significantly.

Naturally, the next question is how does memorization affect the ability of GNNs to learn a desirable function that can generalize well? Yang et al. [66] analyze the training dynamics of GNNs through the lens of Neural Tangent Kernel (NTK) [31, 4] in the over-parametrized regime and propose the interplay of three alignment matrices, namely (1) the alignment between the graph structure ($\mathbf{A}$) and the optimal kernel matrix ($\mathbf{\Theta}^* = \bar{\mathbf{Y}}\bar{\mathbf{Y}}^T$ [19] see Definition 4) denoted by $\mathcal{A}(\mathbf{A}, \mathbf{\Theta}^*)$, which encodes the notion of homophily of the graph, (2) $\mathcal{A}(\mathbf{\Theta_t}, \mathbf{\Theta}^*)$, the *kernel-target alignment* which induces better generalization capabilities especially in learning kernel functions and finally (3) $\mathcal{A}(\mathbf{\Theta_t}, \mathbf{A})$, the *kernel-graph alignment* which allows the NTK matrix to naturally incorporate the adjacency matrix in a node level graph NTK formulation (NL-GNTK). For a $l$ layered GNN with weights $W$ and nodes $x, \tilde{x}$ and the adjacency matrix $\mathbf{A}$, the GNTK is given by

$$\mathbf{\Theta}_t^l(x, \tilde{x}; \mathbf{A}) = \nabla_W f(x; \mathbf{A})^T \nabla_W f(\tilde{x}; \mathbf{A}), \qquad (3)$$

where, $\mathbf{\Theta}_t^l(x, \tilde{x}; \mathbf{A})$ is the NTK matrix that can be seen as a measure of similarity between nodes $x, \tilde{x}$, guided by the adjacency matrix $\mathbf{A}$. Following this line of reasoning, we can see our results in Proposition 1 as an inverse relation between $\mathcal{A}(\mathbf{A}, \mathbf{\Theta}^*)$ and memorization rate. This connection gives us a nice ingress to study the influence of memorization on the training dynamics of GNNs. With this preliminary setup, we now introduce our results that connect memorization rate to the way GNNs learn generalizable functions.

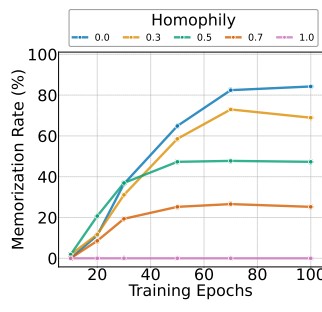

(a) Memorization rate over training epochs.

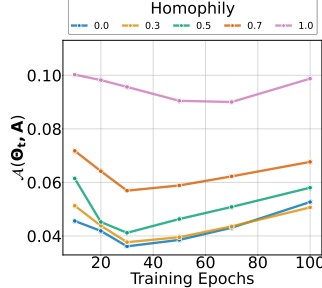

(b) The kernel-graph alignment $\mathcal{A}(\mathbf{\Theta_t}, \mathbf{A})$ over training epochs.

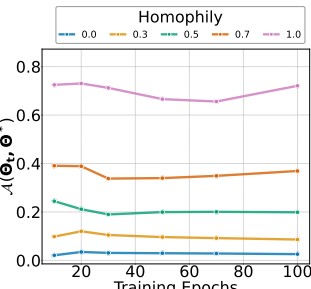

(c) The kernel-target alignment $\mathcal{A}(\mathbf{\Theta_t}, \mathbf{\Theta}^*)$ over training epochs.

Figure 4: **Evolution of the NTK matrix $\mathbf{\Theta_t}$ and its alignment with the adjacency matrix A and the optimal kernel $\mathbf{\Theta}^*$ during training for `syn-cora`, trained on GCN.** (a) Memorization rate is increasing during training, and it is higher for lower homophily graphs. (b) The kernel-graph alignment $\mathcal{A}(\mathbf{\Theta_t}, \mathbf{A})$ improves during training regardless of the graph homophily level. (c) The kernel-target alignment $\mathcal{A}(\mathbf{\Theta_t}, \mathbf{\Theta}^*)$ is poor for low-homophilic graphs.

**Proposition 2 (Implicit bias towards graph structure and memorization.)** *The optimization dynamics of GNNs on low-homophily graphs ($\mathcal{A}(\mathbf{A}, \mathbf{\Theta}^*) \downarrow$) demonstrate a tendency of the GNN model to overfit to the graph structure. For a heterophilic graph (i.e., $h = 0.0$), as training progresses we see that: (i) the memorization rate increases over epochs ($\uparrow$); (ii) the kernel-target alignment $\mathcal{A}(\mathbf{\Theta_t}, \mathbf{\Theta}^*)$ remains low ($\downarrow$), indicating poor generalization; yet (iii) the kernel-graph alignment $\mathcal{A}(\mathbf{\Theta_t}, \mathbf{A})$ tends to increase ($\uparrow$), suggesting the inherent capability of GNNs to leverage the graph structure even when it is detrimental to the task.*

We track the temporal dynamics of the NTK matrix and its alignment with the other kernels as proposed in [66] over training epochs on GCN trained on the `syn-cora` dataset, similar to the setup in Proposition 1. The results are shown in Figure 4. We track the memorization rate of the candidate nodes trained on model $f$ across training epochs; correspondingly, we also track the $\mathcal{A}(\mathbf{\Theta_t}, \mathbf{\Theta}^*)$ and $\mathcal{A}(\mathbf{\Theta_t}, \mathbf{A})$ across epochs. The lines are color coded by the homophily level. Figure 4(a) reveals that the memorization rate consistently increases with training epochs. This rate is higher for low homophilic graphs. In Figure 4(b), we can observe that the kernel-graph alignment tends to improve over time regardless of the homophily level. This reinforces the idea of GNNs leveraging the graph structure during optimization regardless of the homophily level (and thus, whether the graph structure is actually useful for the learning task). This alignment trend contrasts with $\mathcal{A}(\mathbf{\Theta_t}, \mathbf{\Theta}^*)$ behavior (Figure 4(c)), where lower homophily graphs show substantially poorer alignment, confirming the model's inability to learn generalizable functions in these settings. We provide a simple proof in Appendix D.2 to show that a model that shows better kernel-graph alignment cannot simultaneously also obtain better kernel-target alignment in low homophily settings. The interplay of these evolving metrics suggests that the final state of these alignments will be indicative of the overall memorization. To quantify this relationship at convergence, our next proposition presents a correlation analysis between memorization rate and the final kernel matrix alignments.

**Proposition 3 (Negative correlation between MR, $\mathcal{A}(\mathbf{\Theta_t}, \mathbf{\Theta}^*)$ and $\mathcal{A}(\mathbf{\Theta_t}, \mathbf{A})$ alignments.)** *Let $f$ be the GNN model trained for node classification via gradient descent on a graph $\mathcal{G} = (\mathbf{X}, \mathbf{A})$ with labels $\mathbf{Y}$, converging to final weights $W_f$. Let $\mathbf{\Theta}_{final} \in \mathbb{R}^{n \times n}$ be the empirical Neural Tangent Kernel (NTK) corresponding to $W_f$, and $\mathbf{\Theta}^*$ is the optimal kernel. As the kernel-target alignment $\mathcal{A}(\mathbf{\Theta_t}, \mathbf{\Theta}^*)$ improves, the memorization rate decreases. Similarly, as the kernel-graph alignment $\mathcal{A}(\mathbf{\Theta_t}, \mathbf{A})$ improves, the memorization rate decreases.*

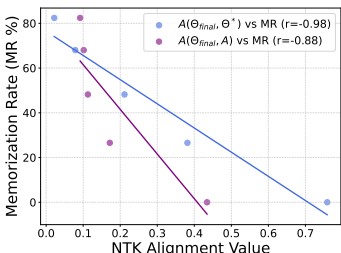

Figure 5: **The inverse relationship between MR and the $\mathcal{A}(\mathbf{\Theta_{final}}, \mathbf{\Theta}^*)$, $\mathcal{A}(\mathbf{\Theta_{final}}, \mathbf{A})$ alignments.**

We measure the correlation between the memorization rate and alignment matrices by setting $\mathbf{\Theta}_t = \mathbf{\Theta}_{final}$, *i.e.,* we take the converged GCN model $f$ trained on the `syn-cora` dataset and use the final weights to calculate the alignment matrices $\mathcal{A}(\mathbf{\Theta}_{final}, \mathbf{A})$ and $\mathcal{A}(\mathbf{\Theta}_{final}, \mathbf{\Theta}^*)$. The results are presented in Figure 5, where we see a Pearson correlation of $r = -0.98$ with respect to the kernel-target alignment ($\mathcal{A}(\mathbf{\Theta}_{final}, \mathbf{\Theta}^*)$) and a correlation coefficient of $r = -0.88$ with respect to the kernel-graph alignment ($\mathcal{A}(\mathbf{\Theta}_{final}, \mathbf{A})$). As the kernel-target alignment $\mathcal{A}(\mathbf{\Theta}_{final}, \mathbf{\Theta}^*)$ improves, which suggests the model's $\mathbf{\Theta}_{final}$ is aligning well with the optimal kernel $\mathbf{\Theta}^*$ leading to good generalization behavior, the need for memorization is obviated. Similarly, when the kernel-graph alignment $\mathcal{A}(\mathbf{\Theta}_{final}, \mathbf{A})$ improves, this suggests that the model is leveraging the graph structure which is obviously useful (because of high homophily) for the learning task, and consequently we see a decrease in the memorization rate.

### 4.3 Feature Space Label Inconsistency is a Strong Indicator of Node-Level Memorization

Our preceding analyses have established a crucial link between graph-level properties like homophily and the overall memorization rate (Proposition 1). We have also analyzed the training dynamics (Proposition 2, Proposition 3) to explain how GNNs' reliance on graph structure can lead to memorization. However, these perspectives do not fully address a fundamental question: why are only specific nodes within a graph highly susceptible to being memorized, while others are learned in a more generalizable manner? Drawing an analogy from DNNs where *atypical* examples are often memorized, we hypothesize that, in GNNs, nodes exhibiting strong discordance between their labels

and their local feature-space context are particularly susceptible to memorization. We formalize this intuition in a proposition below.

**Proposition 4 (Higher label disagreement indicates susceptibility to memorization.)** *Nodes that exhibit a high degree of label inconsistency with their neighbors in feature space are more likely to be memorized by a GNN.*

To quantify this label inconsistency, we introduce the **Label Disagreement Score (LDS)**. Let $S_C$ be the set of candidate nodes and $S_{\text{train\_f}} = S_S \cup S_C$ be the set of nodes used to train model $f$, whose features form the search space for neighbors. For a target node $v_i \in S_C$ with features $x_i \in \mathbb{R}^d$ and label $y_i$, we identify its $k$-nearest neighbors in feature space from $S_{\text{train\_f}}$. The node $v_i$ itself is omitted from its own neighborhood set $N_k(v_i)$; including it would artificially decrease the disagreement score (as $\mathbb{I}[y_i \neq y_i] = 0$) without reflecting the actual inconsistency with its distinct surroundings. Therefore, $N_k(v_i)$ contains $k$ nodes $v_j \in S_{\text{train\_f}} \setminus \{v_i\}$ that minimize the $L_2$ distance $\|x_i - x_j\|_2$. The LDS for node $v_i$ is then defined as:

Table 1: **LDS for $S_C$ on** `syn-cora-h0.0`.

| MemNodes | Non-MemNodes |
|---|---|
| LDS | LDS |
| 0.6270±0.0038 | 0.5187±0.0138 |

$$\text{LDS}_k(v_i) = \frac{1}{k} \sum_{v_j \in N_k(v_i)} \mathbb{I}[y_j \neq y_i] \tag{4}$$

where $\mathbb{I}[\cdot]$ is the indicator function, and $k$ is a chosen hyperparameter (e.g., $k = 3$). The LDS intuitively measures the local *anomaly* in the label-feature manifold. If a node's features place it close to neighbors with predominantly different labels, it presents a conflicting signal to the GNN. To minimize training loss on such a node, the GNN might resort to memorizing it rather than learning a feature transformation that adequately classifies both this target node and its disagreeing neighbors under a shared rule. Following our experimental setup, in Table 1, we measure the LDS for memorized versus non-memorized nodes within the candidate set of the `syn-cora-h0.0` dataset, using a GCN model and averaging over three random seeds. We consistently observe that memorized nodes exhibit, on average, significantly higher LDS than non-memorized nodes. This finding supports our hypothesis that strong conflicting signals arising from high label inconsistency in a node's feature-space neighborhood make that node particularly vulnerable to being memorized by the GNN.

## 5 Extending Evaluation to Real-World Data and Other Tasks

In this section, we present results on memorization scores on real-world datasets. We use GCN [37], GraphSAGE [29] and GATv2 [13] with 3 layers as our backbone GNN models. We also report additional results on larger datasets in Appendix E.3, with Graph Transformer in Appendix E.4 and SGC [65] in Appendix E.5 backbones. The datasets we use are: Cora [41], Citeseer [56], Pubmed [45], which are homophilic datasets. Cornell, Texas, Wisconsin [48], Chameleon, Squirrel [53], and Actor are heterophilic datasets. We divide our datasets into 60%/20%/20% as training, testing, and validation sets, respectively. Out of the 60% training set $S$, we further randomly divide the datasets into three disjoint subsets with ratios $S_S = 50\%$, $S_C = 25\%$, $S_I = 25\%$. We instantiate two GNN models $f$ and $g$, which are trained on $S_S \cup S_C$ and $S_S \cup S_I$, respectively. We train both models on three random seeds. We choose the best model based on the validation accuracy to calculate the memorization scores for the distribution plots, to ensure our memorization results are not an artifact of poor training. Based on Equation (1), the resulting memorization score ranges between -1 and 1, where 0 means no memorization, 1 means the strongest memorization phenomenon on model $f$, and -1 denotes the strongest memorization on $g$.

### 5.1 Memorization Results on Real-World Datasets

**Inverse Relation Between Memorization Rate and Homophily on Real-World Datasets.** Figure 6 shows the distribution of memorization scores for candidate nodes on four real-world datasets trained on a GCN, revealing a similar trend as in Figure 2. That is, candidate nodes of homophilic graphs such as Cora and Citeseer show lower memorization scores than heterophilic graphs such as Chameleon and Squirrel. We also report the average memorization scores and memorization rate along with additional results for different GNN models in Appendix E.

**Evolution of Alignment Matrices on Real-World Graphs.** We verify the results outlined in Section 4 on real-world datasets. We use GCN for all of our experiments. In Figure 7 we show the

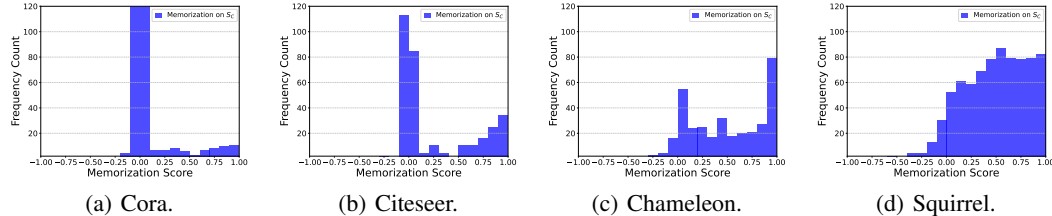

(a) Cora.  (b) Citeseer.  (c) Chameleon.  (d) Squirrel.

Figure 6: **Memorization scores of $S_C$ for Cora, Citeseer, Chameleon, and Squirrel datasets.** The candidate nodes of homophilic graphs exhibit lower memorization scores than heterophilic graphs.

evolution of the alignment matrices on 9 real-world graphs. We can observe in Figure 7(a) that our insights from the synthetic dataset holds true for real-world settings: homophilic graphs such as Cora, Citeseer, and Pubmed exhibit lower memorization compared to the heterophilic graphs like Cornell, Texas, Wisconsin, Chameleon, Squirrel, and Actor. Correspondingly, in Figure 7(b), we can see the kernel-graph alignment $\mathcal{A}(\Theta_t, A)$ improves over training epochs for all the datasets, regardless of the homophily level, highlighting the implicit bias of GNNs [66]; and Figure 7(c) shows that the homophilic graphs have higher kernel-target alignment ($\mathcal{A}(\Theta_t, \Theta^*)$) compared to heterophilic graphs and thus lead to better generalization. These real-world results align well with the trends described in Proposition 2.

Moreover, our findings complement Bechler-Speicher et al. [9] but distinguishes label memorization as a separate phenomenon. Specifically, our analysis of training dynamics (Proposition 2) shows that GNNs' reliance on graph structure (increasing $\mathcal{A}(\Theta_t, A)$), particularly in low-homophily settings where this structure is uninformative for label prediction, can compel the model to memorize individual node labels to minimize training loss. This memorization, driven by the need to fit specific labels, is distinct from structural overfitting; while both involve a potentially suboptimal reliance on graph structure, the latter concerns the model's choice of information source, whereas memorization details how specific training instances are fit when signals are conflicting or uninformative.

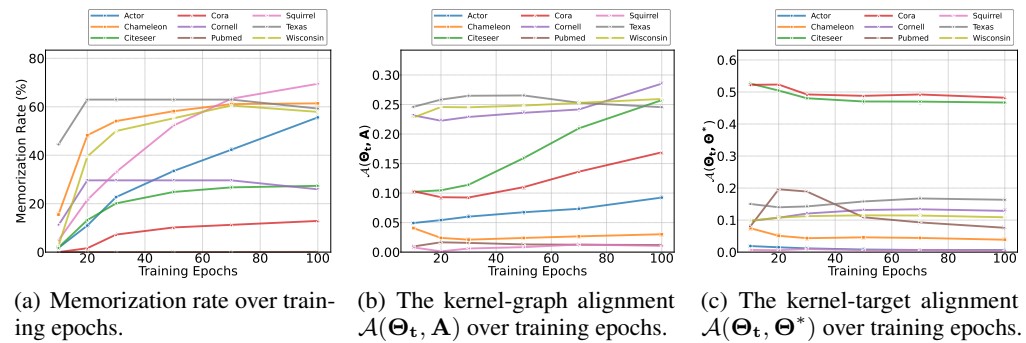

(a) Memorization rate over training epochs.  (b) The kernel-graph alignment $\mathcal{A}(\Theta_t, A)$ over training epochs.  (c) The kernel-target alignment $\mathcal{A}(\Theta_t, \Theta^*)$ over training epochs.

Figure 7: **Evolution of the NTK matrix $\Theta_t$ and its alignment with the adjacency matrix $A$ and the optimal kernel $\Theta^*$ for real-world graphs trained on GCN.** (a) Memorization rate is increasing during training, and it is higher for lower homophily graphs. (b) The kernel-graph alignment $\mathcal{A}(\Theta_t, A)$ improves during training regardless of the graph homophily level. (c) The kernel-target alignment $\mathcal{A}(\Theta_t, \Theta^*)$ is poor for low-homophilic graphs.

**Correlation Between MR, $\mathcal{A}(\Theta_{final}, \Theta^*)$ and $\mathcal{A}(\Theta_{final}, A)$ Alignments on Real-World Graphs.** We also validate Proposition 3 using 8 real-world datasets [3]. In Figure 8, we plot the Pearson correlation coefficients between the memorization rate and the two alignment matrices (*i.e.,* $\mathcal{A}(\Theta_t, \Theta^*)$ and $\mathcal{A}(\Theta_t, A)$). We can observe a negative correlation of $r = -0.79$ between memorization rate and $\mathcal{A}(\Theta_t, \Theta^*)$, and a slightly weaker but nevertheless negative correlation of $r = -0.58$ between memorization rate and $\mathcal{A}(\Theta_t, A)$. These findings indicate that as the kernel-target or kernel-graph alignment improves, the memorization rate tends to decrease. We attribute the relatively weaker

---

[3]We omit Pubmed from this analysis since it exhibits strong resistance to memorization, see Table 5 and Appendix F for discussion.

correlation with $\mathcal{A}(\boldsymbol{\Theta_t}, \mathbf{A})$ to the fact that the actual homophily values of the datasets are highly varied (See Table 18 for the actual values). Nonetheless, the overall trend supports our Proposition 3 on real-world datasets as well.

**Label Disagreement Score for Real-World Datasets.** We calculate LDS for both memorized and non-memorized nodes on the candidate set ($S_C$) averaged over 3 random seeds for Cora, Citeseer, Chameleon and Squirrel datasets in Table 2 along with their statistical significance. From the table, we can see that the memorized nodes have a higher label disagreement scores compared to the non-memorized nodes and this trend is consistent across both homophilic and heterophilic graphs. Thus, we can use this metric to understand why only certain nodes get memorized, a node belonging to a neighborhood where the labels strongly give a conflicting signal for the GNN aggregation step ends up getting memorized by the model. More results on LDS are presented in Appendix F. In the next section, we discuss graph rewiring as a possible strategy to decrease memorization in GNNs, further we also discuss the implications of memorization to privacy risk.

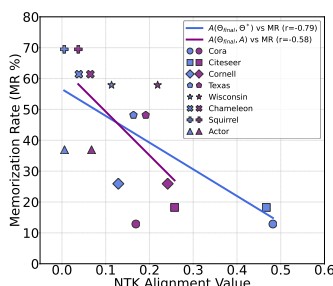

Figure 8: **The inverse relationship between MR and the $\mathcal{A}(\boldsymbol{\Theta_{final}}, \boldsymbol{\Theta^*})$, $\mathcal{A}(\boldsymbol{\Theta_{final}}, \mathbf{A})$ alignments for real-world datasets.**

Table 2: **LDS for memorized nodes (MemNodes) vs non-memorized nodes (Non-MemNodes) in the $S_C$, trained on GCN averaged over 3 random seeds.**

| Dataset | MemNodes LDS | Non-MemNodes LDS | $p$-value ($< 0.01$) | Effect Size |
|---|---|---|---|---|
| Cora | 0.8016±0.0186 | 0.5754±0.0010 | 0.0024 | 14.39 |
| Citeseer | 0.7985±0.0114 | 0.6192±0.0040 | 0.0029 | 13.13 |
| Chameleon | 0.8433±0.0006 | 0.7362±0.0016 | 0.0000 | 113.68 |
| Squirrel | 0.8617±0.0032 | 0.7675±0.0040 | 0.0020 | 15.98 |

## 5.2 Memorization in Graph Classification

We also extend our memorization framework to graph classification. The dataset $S$ now consists of graphs, and memorization is quantified as the change in expected behavior between models $f$ trained on $S$ and models $g$ trained on $S \setminus gr_i$, where $gr_i$ is the graph under test. We evaluate on MUTAG, AIDS, BZR, and COX2 [43] using a GCN backbone. The data is split into 60%/20%/20% for training, testing, and validation, with the training set further divided into shared ($S_S$), candidate ($S_C$), and independent ($S_I$) subsets in 50%/25%/25% ratios. Results (Table 3) indicate that the GCN learns effectively without memorization.

**Feature Noise Drives Memorization.** We hypothesize that this is because there are no *outlier* graphs, making it possible to learn without memorization, following the reasoning of Feldman [21]. To test this hypothesis and assess whether our framework is sensitive to measuring memorization in graph classification tasks when it does appear, we introduce Gaussian noise with $\sigma \in \{0.1, 0.3, 0.5, 0.7, 0.9\}$ to node features of all the nodes of candidate graphs, turning them into outliers, and recompute memorization scores. Our results in Table 3 show that as noise increases, the graphs become more atypical and the framework successfully detects memorization, with MUTAG showing a clear monotonic rise in memorization rate and average score, whereas other datasets exhibit less consistent trends. While further work is needed to understand the mechanisms underlying memorization in graph classification, these findings highlight the versatility of our proposed approach.

Table 3: Average memorization score and memorization rate of candidate graphs with GCN.

| | MUTAG | | AIDS | | BZR | | COX2 | |
|---|---|---|---|---|---|---|---|---|
| Noise Level | Avg MemScore | MR (%) | Avg MemScore | MR (%) | Avg MemScore | MR (%) | Avg MemScore | MR (%) |
| 0.0 | 0.0011±0.0045 | 0 | 0.0767±0.0260 | 0 | -0.0096±0.0067 | 0 | 0.0068±0.0241 | 0 |
| 0.1 | 0.0767±0.0260 | 2.2±1.9 | 0.0767±0.0260 | 0 | 0.1993±0.0137 | 9.2±3.1 | 0.1939±0.0116 | 2.6±1.3 |
| 0.3 | 0.2352±0.0744 | 18.3±9.9 | 0.1351±0.0298 | 6.5±3.2 | 0.2965±0.0064 | 27.7±0.0 | 0.2326±0.0145 | 21.5±1.5 |
| 0.5 | 0.3419±0.0420 | 33.3±3.7 | 0.2086±0.0295 | 13.7±1.2 | 0.2091±0.0109 | 15.9±0.9 | 0.2856±0.0012 | 28.9±0.0 |
| 0.7 | 0.3680±0.0239 | 39.8±4.9 | 0.2465±0.0087 | 18.8±1.4 | 0.1790±0.0010 | 18.5±0.0 | 0.2237±0.0101 | 23.7±0.0 |
| 0.9 | 0.4544±0.0302 | 46.2±6.7 | 0.2367±0.0346 | 17.6±2.2 | 0.1998±0.0006 | 20.0±0.0 | 0.1838±0.0092 | 18.4±0.0 |

# 6 Privacy Risks and the Real-World Impact of Memorization

We finally turn to the real-world impact of memorization. In DNNs, memorization has often been linked to privacy risks for the memorized data points, *e.g.,* [15, 18]. In this section, we highlight that the same holds for GNNs. Based on our findings from the previous sections, we explore graph rewiring as a mitigation strategy. Our findings highlight the effectiveness of this approach for mitigating memorization and reducing the associated privacy risks while maintaining high model performance.

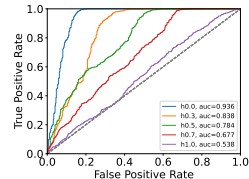 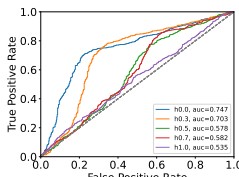

Figure 9: **MIA risk of GNNs trained on original** `syn-cora` **(left) and rewired** `syn-cora` **(right).** `h0.0` `(1.0)` means homophily level 0.0 (1.0).

**Analyzing Privacy Risks.** Membership inference attacks (MIAs) [59] are a standard tool to assess privacy leakage in machine learning. In these attacks, an adversary aims at inferring whether a certain data point was included in the training set of a model. We rely on the Likelihood Ratio Attack (LiRA) [16] to assess the privacy risk in our GNNs. Our results for `syn-cora` are presented in Figure 9 (left), where we measure the Area Under Curve (AUC) to characterize the vulnerability of the GNN model. We also present the TPR (True Positive Rate) at 1% FPR (False Positive Rate) in Table 13, Appendix G. We observe that, nodes in low homophily graphs, such as `syn-cora-h0.0`, which exhibit high memorization, are most susceptible to MIAs with an AUC value of $0.936$, whereas nodes in a highly homophilic graph, *e.g.,* `syn-cora-h1.0`, show a value of $0.538$, close to random guessing. Our results indicate that nodes in lower homophily graphs, have a significant higher risk of leaking from trained GNN models due to memorization. This prompts the question *How can we reduce memorization and its resulting privacy risks?*.

**Graph Rewiring to Prevent Privacy Leakage.** Our Proposition 1 suggests that one of the key contributors to memorization is the graph homophily, which means it should be possible to control the memorization by influencing the homophily level. However, since calculating homophily requires access to all the node labels, it cannot be directly optimized for. Therefore, we adopt a feature similarity-based graph rewiring framework proposed in [54], which aims to maximize the feature similarity (based on cosine similarity) between nodes by adding or deleting edges, indirectly influencing the homophily level. See Appendix G for a full description of the approach. It is assumed that after applying the rewiring method, the graph's homophily level will increase, leading to less memorization and MIA risk.

**Evaluation.** To validate this hypothesis, we rewire the `syn-cora` graphs by adding a specific number of edges to the graph. The homophily level of the rewired `syn-cora` graphs indeed increases. We then train new GNN models on the rewired graphs and analyze the memorization behavior of those models using `NCMemo`. The detailed implementation setups and results are presented in Appendix G. We also perform the LiRA attack again on those models. Our results in Figure 9 (right) show that the MIA risk is significantly reduced compared to the models trained on the original graphs, *i.e.,* the AUC scores have dramatically dropped for all `syn-cora` graphs. The drop is most significant for the low homophily graphs (*e.g.,* `syn-cora-h0.0`), where the AUC scores have dropped by more than $20\%$. This confirms our hypothesis that graph rewiring can be a simple memorization mitigation strategy to prevent the real-world privacy risks for memorized samples. We also consider label smoothing as an alternative memorization mitigation strategy and compare the results with graph rewiring in Appendix G.1.

# 7 Conclusions

In this work, we propose `NCMemo`, the first framework to quantify label memorization in GNNs. We identify both graph-level and node-level factors that influence memorization, such as graph homophily and feature–label inconsistency. We also extend our framework to graph classification and provide promising initial results. Through experiments on real-world privacy risks stemming from GNN memorization, we further demonstrate that graph rewiring can effectively reduce both memorization and its associated privacy risks. Overall, our work advances the understanding of GNN learning dynamics and contributes to their more privacy-preserving deployment.

## Acknowledgments and Disclosure of Funding

We gratefully acknowledge funding from the Deutsche Forschungsgemeinschaft (DFG, German Research Foundation), Project number 550224287.

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

# A    Appendix

**Limitations.** Our work proposes an end-to-end framework to study memorization in semi-supervised node classification setting in GNNs. While our findings on the interplay between homophily, training dynamics, and memorization offer significant insights, this foundational study also presents several avenues for future exploration.

1. **Conceptualizing Memorization on Non-IID Graph Data:** Our approach, inspired by the leave-one-out framework of Feldman [21], quantifies memorization by observing changes in model behavior when a node's label is masked during training. In the semi-supervised node classification setting, where the entire graph structure and all node features are accessible during training, our method correctly isolates the influence of a single training label. The difference in predictions between a model trained with a specific node's label and one without it serves as a robust measure of that label's memorization as shown through our numerous experiments. However, this is a foundational step. The inherent dependencies between nodes in a graph suggest that more intricate definitions of memorization could be developed. Future work could explore how the memorization of one node influences the predictions on its neighbors, moving from a single-node perspective to a more holistic, subgraph-level understanding of memorization. This could lead to new metrics that capture the unique relational aspects of GNNs.

2. **Effectiveness of the Label Disagreement Score (LDS):** Our proposed Label Disagreement Score (LDS) effectively identifies nodes prone to memorization by measuring local inconsistencies of a node in the label-feature space. As demonstrated, we find that nodes with higher LDS are more likely to be memorized. Nevertheless, the paper acknowledges that the effectiveness of LDS might be reduced in highly heterophilic graphs with non-uniform feature and label distributions, such as the Actor dataset. In such complex scenarios, the local neighborhood might not provide a clear signal, making it difficult to distinguish between *atypical* nodes that require memorization and the general noise of a highly diverse environment.

3. **Limitations of Graph Rewiring:** As an initial strategy to mitigate memorization, we explored feature-similarity-based graph rewiring [54]. Our results confirm that this can reduce memorization and associated privacy risks by increasing graph homophily. However, this approach is relatively simple and may not be effective, especially on real-world graphs. The process of adding or removing edges based solely on feature similarity might result in information loss.

**Future work.** Studying memorization for other tasks such as link prediction and graph classification can be an interesting future work. In Section 5.2, we already extend our memorization framework to characterize memorization in graph classification setting. We find interesting connections between node feature noise and the model's ability to memorize entire graphs. Future work can also focus on understanding the role of over-squashing [2] and over-smoothing [35] on memorization. In our work, we have primarily analyzed the training dynamics of GCN, future work could also extend this analysis to GATs [62, 13] and Graph Transformers [51] since the attention mechanism can have non-trivial effects on memorization.

# B    Broad Impact

We propose a new framework to study the memorization phenomenon in GNNs. Our work provides a comprehensive understanding of the factors that contribute to memorization in GNNs, including the role of homophily, label informativeness, kernel alignment, and feature-label inconsistency. We also discuss the implications of memorization for privacy risks. Our findings can help researchers and model developers better understand the memorization phenomenon in GNNs, and encourage the development of more robust and generalizable GNN architectures.

# C    Definitions

In this section we provide definitions of all the metrics we have used.

**Definition 1 (Edge Homophily)** *The edge homophily ratio is defined as the fraction of edges in a graph that connect nodes of the same class and is given by*

$$h = \frac{|\{(u,v) : (u,v) \in \mathcal{E} \wedge y_u = y_v\}|}{|\mathcal{E}|} \tag{5}$$

*This measure is used to generate the synthetic datasets in [68].*

**Definition 2 (Node Label Informativeness [49])** *Given an edge $(u,v)$ sampled from $\mathcal{E}$ and let $y_u$ and $y_v$ be the respective class labels. The label informativeness gives a measure of predicting the label $y_u$, given we know the label $y_v$ and is encoded by the normalized mutual information:*

$$NLI = \frac{H(y_u) - H(y_u|y_v)}{H(y_u)} \tag{6}$$

*where $H(y_u)$ is the entropy that tells us how difficult it would be to predict the label $u$ without knowing $y_v$. A higher label informativeness implies a node's neighborhood gives enough information to predict the label. The label informativeness is computed for all pairs of edges and labels to give a final graph level score.*

**Definition 3 (Homophily in terms of alignment [66])** *The notion of homophily can also be encoded in the form of alignment between the adjacency matrix of the graph $\mathbf{A}$ and the optimal kernel $\Theta^*$ [19], i.e., $\mathcal{A}(\mathbf{A}, \Theta^*)$ measures if two connected nodes have the same labels.*

**Definition 4 (Optimal kernel ($\Theta^*$) and kernel alignment)** *The optimal kernel $\Theta^*$ proposed in [19] is defined as $\Theta^* = \bar{\mathbf{Y}}\bar{\mathbf{Y}}^T$ and it measures how similar two instances of data points are. Further, we can define a similarity metric that measures the alignment between two kernels $\mathbf{K_1}$ and $\mathbf{K_2}$ as*

$$\mathcal{A}(\mathbf{K_1}, \mathbf{K_2}) = \frac{\langle \mathbf{K_1}, \mathbf{K_2} \rangle_F}{||\mathbf{K_1}||_F ||\mathbf{K_2}||_F} \tag{7}$$

*This metric can be seen as a generalization of the cosine similarity applied to matrices and it satisfies the triangle inequality.*

# D   Proof for Proposition 2

In this section, we provide a simple proof to support our empirical results in Proposition 2. We showed that one of the main reasons GNNs exhibit memorization is their failure to reconcile the conflicting signals induced by the training dynamics of GNNs, *i.e.,* the implicit tendency to align the $\Theta_t$ with the $\mathbf{A}$, even when the graph structure is uninformative (*i.e.,* exhibits low $\mathcal{A}(\mathbf{A}, \Theta^*)$) and can be detrimental to the learning task, while, the overall training objective remains minimizing the training loss. We adopt the terminologies from [66].

## D.1   Setup and Assumptions

We consider a GNN trained for node classification on graph $\mathcal{G} = (\mathcal{V}, \mathcal{E})$ with propagation matrix $\mathbf{A}$. Let $\mathcal{D}_{\text{train}} = \{(v_i, y_i)\}_{v_i \in \mathcal{V}_{\text{train}}}$ be the labeled training nodes ($n_l = |\mathcal{V}_{\text{train}}|$). Let $\mathbf{F}_t \in \mathbb{R}^{n_l}$ be predictions at time $t$ and $\mathbf{Y} \in \mathbb{R}^{n_l}$ be true labels. We minimize the empirical risk $\mathcal{L}(\mathbf{F}_t, \mathbf{Y}) = \frac{1}{2}||\mathbf{F}_t - \mathbf{Y}||^2$.

**Assumption 1 (GNN training dynamics increase kernel-graph alignment)** *The optimization process for GNNs implicitly leverages the graph structure $\mathbf{A}$, such that $\mathcal{A}(\Theta_t, \mathbf{A})$ tends to increase during training.*

**Assumption 2 (Low homophily)** *In low homophily settings, alignment between the $\mathbf{A}$ and $\Theta^*$ is low.*

## D.2 Low Homophily ($\mathcal{A}(\mathbf{A}, \Theta^*)$) Implies Low Kernel-Target Alignment ($\mathcal{A}(\Theta_t, \Theta^*)$)

The alignment between two kernels [19, 66] $\mathbf{K_1}$ and $\mathbf{K_2}$ can be characterized by the cosine similarity as shown in Equation 7. Let $\phi(\mathbf{K_1}, \mathbf{K_2}) = \arccos(\mathcal{A}(\mathbf{K_1}, \mathbf{K_2}))$ denote the angle between the kernel matrices. Our proof relies on the simple triangle inequality and a geometric approach to show that, a GNN that aligns its adjacency matrix $\mathbf{A}$ with the neural tangent kernel $\Theta_t$ cannot simultaneously also align well with the $\Theta^*$ and thus leads to memorization.

*Proof:*

We define three angles $\epsilon$, $\delta$ and $\gamma$ as the angles between the matrices $\mathcal{A}(\Theta_t, \mathbf{A})$, $\mathcal{A}(\mathbf{A}, \Theta^*)$ and $\mathcal{A}(\Theta_t, \Theta^*)$ respectively, that is

- $\epsilon = \arccos(\mathcal{A}(\Theta_t, \mathbf{A}))$ be the angle between $\Theta_t$ and $\mathbf{A}$.
- $\delta = \arccos(\mathcal{A}(\mathbf{A}, \Theta^*))$ be the angle between $\mathbf{A}$ and $\Theta^*$.
- $\gamma = \arccos(\mathcal{A}(\Theta_t, \Theta^*))$ be the angle between $\Theta_t$ and $\Theta^*$.

Using the triangle inequality on the angles we write two inequalities as

$$\gamma \leq \epsilon + \delta \tag{8}$$

$$\delta \leq \epsilon + \gamma \tag{9}$$

Rearranging 9 to establish a *lower bound* on $\gamma$:

$$\gamma \geq \delta - \epsilon \tag{10}$$

Analyzing the inequality 10 in light of our assumptions we get

1. Based on Assumption 1, a high kernel-graph alignment implies, the angle $\epsilon$ between $\mathbf{A}$ and $\Theta_t$ should be small (ideally, $0°$), from a geometrical perspective we can think of this as two vectors pointing in the same direction[4]

2. Based on Assumption 2, in low homophily settings, the angle $\delta$ between $\mathbf{A}$ and $\Theta^*$ should be large (e.g, $90°$)

Therefore, the term $\delta - \epsilon$ represents a large angle minus a small angle, which is still a large angle (quantitatively close to $\delta$). Inequality 10 states that $\gamma$ must be greater than or equal to this large angle ($\delta - \epsilon$). This forces $\gamma$ itself to be large.

A large angle $\gamma$ between the kernel $\Theta_t$ and the optimal kernel $\Theta^*$ signifies a poor alignment between them. Since $\gamma = \arccos(\mathcal{A}(\Theta_t, \Theta^*))$ and the arccos function is monotonically decreasing on its domain $[-1, 1]$, a large angle $\gamma$ (specifically, $\gamma \geq \delta - \epsilon$, where $\delta - \epsilon$ is close to $90°$ or more) necessarily implies that the alignment value $\mathcal{A}(\Theta_t, \Theta^*)$ must be low (close to 0 or negative).

Combining the upper bound equation 8 and the lower bound equation 10, we get $\delta - \epsilon \leq \gamma \leq \delta + \epsilon$. This confirms that when $\epsilon$ is small, $\gamma$ is tightly constrained around the large value $\delta$ ($\gamma \approx \delta$).

Therefore, the geometric constraints imposed by the triangle inequality demonstrate that high kernel-graph alignment (small $\epsilon$) under low homophily (large $\delta$) forces the angle $\gamma$ between the learned kernel and the optimal kernel to be large, resulting in low kernel-target alignment ($\mathcal{A}(\Theta_t, \Theta^*)$).

$\square$

We illustrate our proof through a conceptual diagram in Figure 10. We visualize a hypothetical loss landscape of a GNN model (*e.g.,* GCN trained on `syn-cora` as in Proposition 2) in a 2D contour plot. The blue color represents low loss regions with wider basins, indicating global minima that favor generalization. In contrast, the narrower, red regions correspond to local minima that may achieve low training loss but typically lead to memorization and poor test performance. We visualize the alignment matrices as vectors in this 2D space. We get two regimes:

---

[4]Note that, these are alignment matrices and not simple vectors, besides, these kernels exist in high dimensional space which is not feasible to visualize. Our intent here is to provide an intuitive picture of how memorization emerges in GNNs.

**Low homophily regime -** We know that GNNs possess an implicit bias to leverage the graph structure during optimization, leading to kernel-graph alignment, $\mathcal{A}(\Theta_t, \mathbf{A})$ [66]. This tendency is further supported by our experiments in Proposition 2, which demonstrate that kernel-graph alignment increases as training progresses. However, whether such alignment benefits the model depends critically on the graph homophily level, which is encoded in the alignment between the adjacency matrix $\mathbf{A}$ and the optimal kernel $\Theta^*$, denoted as $\mathcal{A}(\mathbf{A}, \Theta^*)$. Both our geometrical proof and empirical results suggest that in low homophily regimes, the model's tendency to align its NTK matrix with the adjacency matrix becomes problematic. Even when the graph structure contains little task-relevant information, this alignment bias pulls the model away from the optimal kernel $\Theta^*$, which would otherwise guide the model toward better generalization. We illustrate this phenomenon conceptually in Figure 10(a), where we visualize the adjacency matrix $\mathbf{A}$, the NTK kernel matrix $\Theta_t$, and the optimal kernel matrix $\Theta^*$ as 1D vectors in a hypothetical 2D loss landscape, represented in red, green, and blue colors, respectively. In this visualization, the NTK kernel matrix $\Theta_t$ aligns closely with $\mathbf{A}$, as indicated by the small angle $\epsilon$ between them (see Appendix D.2), yet it points in a direction far from the $\Theta^*$, which represents a generalization minimum. The purple trajectory traces the path the model actually follows during optimization, resulting in a biased solution that achieves low training loss through memorization but generalizes poorly to unseen data.

**High homophily regime -** In contrast, within the high homophily regime, the increasing kernel-graph alignment $\mathcal{A}(\Theta_t, \mathbf{A})$, *i.e.,* the model's tendency to leverage the graph structure, becomes highly beneficial for the learning task. We demonstrate this in Figure 10(b), where, as before, we visualize the adjacency matrix $\mathbf{A}$, the NTK kernel matrix $\Theta_t$, and the optimal kernel matrix $\Theta^*$ as 1D vectors in a hypothetical 2D loss landscape, shown in red, green, and blue colors, respectively. In this scenario, we observe that the alignment between $\Theta_t$ and the adjacency matrix $\mathbf{A}$ does not misguide the model away from the direction of the optimal kernel $\Theta^*$. Instead, this alignment actually serves as a beneficial guide that helps the model toward a solution close to the optimal kernel. Consequently, the model converges to a solution that generalizes well, performing well on both training and testing data.

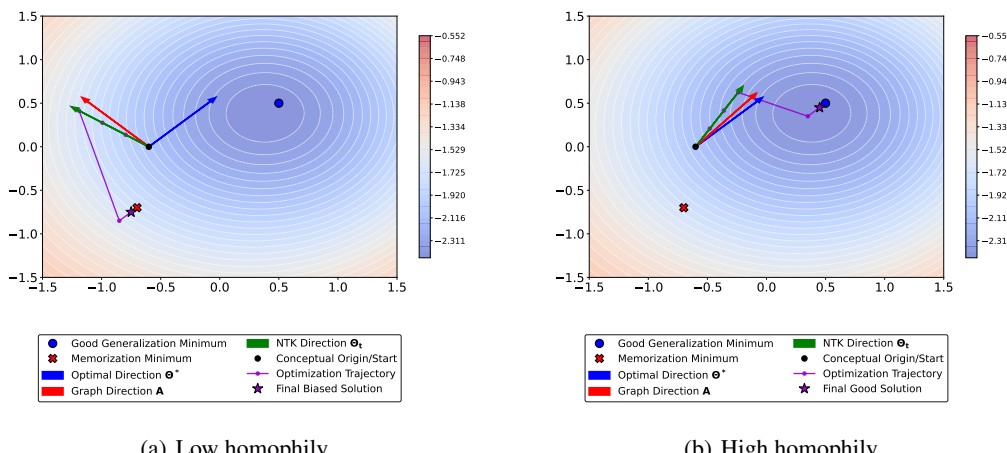

(a) Low homophily.        (b) High homophily.

Figure 10: **We illustrate the emergence of memorization in this conceptual diagram.** We visualize the alignment matrices as 1D vectors on a hypothetical 2D loss landscape of some GNN model, for instance a GCN trained on `syn-cora` dataset as in Proposition 2. In low homophily regimes, the model's tendency to align the $\mathbf{A}$ with $\Theta_t$ (kernel-graph alignment) results in the model deviating away from the optimal kernel $\Theta^*$ resulting in memorization. Contrarily, in high homophily regimes, the kernel-graph alignment does not induce an opposing behavior with respect to the optimal kernel, leading to good generalization.

## D.3 Notations

| Symbol | Description |
|--------|-------------|
| $G = (A, X)$ | A graph with an adjacency matrix $A$ and a node feature matrix $X$. |
| $\mathcal{D}$ | The true, underlying data distribution over node-label pairs $(v, y)$. |
| $\mathcal{D}_{\text{train}}$ | The training set, a collection of $n$ labeled nodes $\{(v_i, y_i)\}_{i=1}^n$ sampled from $\mathcal{D}$. |
| $v_i, v_j$ | A specific node in the graph, indexed by $i$ or $j$. |
| $y_i, y_j$ | The ground-truth label for node $v_i$ or $v_j$. |
| $\mathcal{T}$ | The learning algorithm (e.g., GNN training via gradient descent). |
| $f$ | A predictive function or model learned by the algorithm $\mathcal{T}$. |
| $f_{\mathcal{D}_{\text{train}}}$ | The specific model function learned by algorithm $\mathcal{A}$ when trained on the set $\mathcal{D}_{\text{train}}$. |
| $f_{\mathcal{D}_{\text{train}} \setminus \{v_i\}}$ | The leave-one-out model, trained on $\mathcal{D}_{\text{train}}$ after excluding the $i$-th sample $(v_i, y_i)$. |
| $\mathbb{P}[\cdot]$ | Denotes the probability of an event. The subscript indicates the source of randomness. |
| $\mathbb{E}[[]\cdot]$ | Denotes the expectation of a random variable. |
| $\text{mem}(\mathcal{T}, \mathcal{D}_{\text{train}}, i)$ | The memorization score for the $i$-th sample in the training set $\mathcal{D}_{\text{train}}$. |
| $\mathcal{E}_{\text{gen}}(f)$ | The generalization error of a model $f$, its expected error on the true distribution $\mathcal{D}$. |
| $\mathcal{E}_{\text{train}}(f, \mathcal{D}_{\text{train}})$ | The empirical training error of a model $f$ on the specific set $\mathcal{D}_{\text{train}}$. |

## D.4 Detailed Proof for Proposition 2

In this section, we leverage the results (Lemma 4.2 and 4.3) proposed by Feldman [21] to highlight two regimes of learning conditioned on the graph homophily.

### D.4.1 High-homophily regime

GNNs have an implicit bias to leverage the graph structure during optimization. When the input graph has high homophily, the GNN's inductive bias to aggregate messages from the neighbors aligns well with the data structure (graph whose neighboring nodes have similar labels) in such cases a GNN can obtain good generalization performance.

**Lemma 1** *(Lemma 4.2 Feldman [21]) For a given dataset $S$, a learning algorithm $\mathcal{T}$ and a distribution $P$ over the input-output pair $X \times Y$, the memorization score is related to the generalization gap as follows*

$$\frac{1}{n} \mathbb{E}_{\mathcal{S} \sim P^n} \left[ \sum_{i \in [n]} \text{mem}(\mathcal{T}, S, i) \right] = \mathbb{E}[\mathcal{E}_{\text{train}}] - \mathbb{E}[\mathcal{E}_{\text{gen}}] \tag{11}$$

In high-homophily graphs, the generalization gap is low, this is also supported by our extensive experiments on synthetic graphs and real-world graphs and also our NTK analysis. Thus, we have

$$\mathbb{E}[\mathcal{E}_{\text{train}}] - \mathbb{E}[\mathcal{E}_{\text{gen}}] \approx 0 \implies \text{mem}(\mathcal{T}, S, i) \approx 0 \tag{12}$$

In practice, our experiments reveal even in high-homophily graphs, a small number of nodes are memorized, however the cost of not fitting these nodes does not incur a high penalty on the training error [21].

### D.4.2 Low-homophily regime

Explaining memorization in this regime forms a central theme of our work. As seen earlier in Proposition 2, in low-homophily graphs the graph structure is not informative but the tendency of the

GNNs to still leverage the graph structure results in a model that has to memorize to achieve zero training loss. This is explained by Feldman [21]'s Lemma 4.3 which we adapt to our GNN setting as follows

**Lemma 2** *(Lemma 4.3, Feldman [21]) The empirical error on a high LDS node $v_i$ is related to the leave-one-out error. For an atypical node $v_i$ with a high Label Disagreement Score (LDS), the model trained without it, $f_{\mathcal{S}\setminus\{v_i\}}$, relies on an uninformative graph structure, resulting in a high leave-one-out error. The relation between leave-one-out-error, memorization score and the model's empirical error is given by*

$$\mathbb{P}_{f\sim\mathcal{T}(\mathcal{S})}[f(v_i)\neq y_i] = \mathbb{P}_{f\sim\mathcal{A}(\mathcal{S}\setminus\{v_i\})}[f(v_i)\neq y_i] - \texttt{mem}(\mathcal{T},\mathcal{S},i) \qquad (13)$$

where, the leave-one-out-error is high because the model needs to see the label of the node $v_i$ with high LDS to correctly predict it.

$$\mathbb{P}_{f\sim\mathcal{T}(S\setminus\{v_i\})}[f(v_i)\neq y_i] \to 1 \qquad (14)$$

However, the overall training objective strives to achieve zero empirical error

$$\mathbb{P}_{f\sim\mathcal{T}(S)}[f(v_i)\neq y_i] = 0 \qquad (15)$$

Substituting Equations 14 and 15 in 13 we get,

$$0 \approx 1 - \texttt{mem}(\mathcal{T},\mathcal{S},i) \implies \texttt{mem}(\mathcal{T},\mathcal{S},i) \approx 1 \qquad (16)$$

Thus, in low-homophily settings the atypical nodes characterized by our Label Disagreement Score need to be memorized by the model in order to achieve zero training loss.

# E Additional Results on Memorization Scores for Node Classification

In this section, we present additional memorization results on both the real-world and synthetic datasets.

## E.1 Our Memorization Score is Well Behaved

**Synthetic dataset** In Figure 11, we plot the memorization scores vs frequency for `syn-cora` across the data categories ($S_C, S_S, S_I, S_E$). We can see that the candidate nodes show higher memorization scores compared to the other category nodes such as $S_S$ or $S_E$. We also present the average memorization scores (with 95% confidence interval (CI)) and the memorization rate of $S_C$ in Table 4 for `syn-cora` dataset, trained on GCN model.

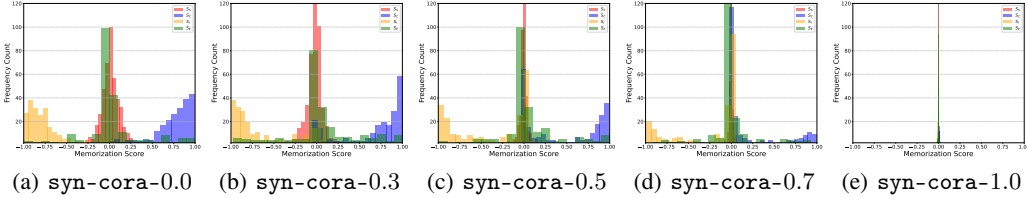

(a) `syn-cora`-0.0    (b) `syn-cora`-0.3    (c) `syn-cora`-0.5    (d) `syn-cora`-0.7    (e) `syn-cora`-1.0

Figure 11: Distribution of Memorization Scores on `syn-cora`: Comparison of memorization rates across node categories for `syn-cora` graphs with increasing homophily levels trained with GCN. GNNs exhibits memorization behavior, *i.e.,* the memorization scores are higher (lower) for $S_C$ ($S_I$) compared to $S_S$ or $S_E$.

**Real-world graphs.** Similarly, Figure 12, Figure 13, and Figure 14 show the distribution of memorization scores for nine real-world datasets trained on GCN, GATv2, and GraphSAGE, respectively.

Table 4: Memorization score (MemScore) and Memorization rate (MR) for the candidate nodes in `syn-cora` dataset averaged over 3 random seeds on a GCN. Heterophilic graphs have a higher memorization score and rate than homophilic graphs.

| Dataset | Avg MemScore | MR (%) |
|---|---|---|
| syn-cora-h0.0 | 0.7091±0.0168 | 83.63±1.82 |
| syn-cora-h0.3 | 0.5573±0.0047 | 63.96±1.19 |
| syn-cora-h0.5 | 0.4106±0.0126 | 45.95±1.62 |
| syn-cora-h0.7 | 0.2616±0.0070 | 25.98±1.45 |
| syn-cora-h1.0 | 0.0021±0.0005 | 0.00±0.00 |

It can be observed that for most of graphs, the nodes in $S_C(S_I)$ have higher (lower) memorization scores than nodes in $S_E$ and $S_S$. We also present the average memorization scores and the memorization rate of $S_C$ in Table 5 for 9 datasets trained on different GNN models. We observe that heterophilic graphs, such as Cornell, Texas, Wisconsin, Chameleon, Squirrel and Actor, have a higher memorization score and memorization rate than the homophilic graphs, *i.e.,* Cora, Citeseer and Pubmed, which is consistent with our Proposition 1. We can observe from Table 5 that Pubmed shows the most resistance to memorization as it has, a very low average memorization score. We can explain this behavior by invoking our Label Disagreement Score reported in Table 10, where we can see the LDS values for both memorized and non-memorized nodes are very similar, indicating that Pubmed has a highly informative neighborhood that can be leveraged by the model to learn generalizable patterns. To formally verify that nodes in $S_C$ ($S_I$) have statistically significantly higher (lower) memorization scores than nodes in $S_S$ and $S_E$, we also conduct a t-test on the memorization scores of all nodes in each category. The details of which are outlined in the Appendix E.2.

Table 5: Memorization score (MemScore) and Memorization rate (MR) for the candidate nodes on 9 real-world dataset, averaged over 3 random seeds. Heterophilic graphs have a higher memorization score and rate than homophilic graphs.

| | Cora | | Citeseer | | Pubmed | |
|---|---|---|---|---|---|---|
| Model | Avg MemScore | MR (%) | Avg MemScore | MR (%) | Avg MemScore | MR (%) |
| GCN | 0.1033±0.0006 | 9.7±0.1 | 0.2661±0.0048 | 28.4±0.8 | 0.0941±0.0012 | 8.1±0.2 |
| GATv2 | 0.1663±0.0165 | 17.0±2.0 | 0.2621±0.0114 | 26.3±0.9 | 0.1312±0.0044 | 12.9±0.5 |
| GraphSAGE | 0.1029±0.0041 | 10.0±0.4 | 0.2411±0.0019 | 24.3±0.6 | 0.0973±0.0023 | 9.4±0.3 |

| | Cornell | | Texas | | Wisconsin | |
|---|---|---|---|---|---|---|
| Model | Avg MemScore | MR (%) | Avg MemScore | MR (%) | Avg MemScore | MR (%) |
| GCN | 0.3695±0.0153 | 36.5±1.7 | 0.3257±0.0107 | 31.1±1.2 | 0.2873±0.0116 | 29.2±0.9 |
| GATv2 | 0.4102±0.0089 | 39.0±1.0 | 0.3701±0.0111 | 35.4±1.4 | 0.3227±0.0135 | 32.0±1.3 |
| GraphSAGE | 0.3542±0.0130 | 35.0±1.6 | 0.3312±0.0087 | 33.0±1.1 | 0.3004±0.0120 | 30.1±0.9 |

| | Chameleon | | Squirrel | | Actor | |
|---|---|---|---|---|---|---|
| Model | Avg MemScore | MR (%) | Avg MemScore | MR (%) | Avg MemScore | MR (%) |
| GCN | 0.5285±0.0053 | 52.9±0.3 | 0.5050±0.0133 | 53.6±1.3 | 0.3821±0.0087 | 38.3±0.6 |
| GATv2 | 0.4962±0.0269 | 50.3±3.3 | 0.5131±0.0179 | 52.9±1.9 | 0.3599±0.0113 | 35.9±1.0 |
| GraphSAGE | 0.4570±0.0020 | 46.0±0.4 | 0.5372±0.0021 | 54.5±1.1 | 0.3715±0.0071 | 37.0±0.5 |

## E.2 Statistical Significance of Memorization Scores

In this section, we present the results for t-tests conducted on the memorization scores obtained from training GCN on real-world datasets in Table 6. We test the null hypothesis $\mathcal{H}_0 := \mathcal{M}(S_C) \leq \mathcal{M}(S_S)$. Rejecting the null hypothesis indicates that the memorization scores of nodes in $S_C$ are significantly higher than those in $S_S$. Our results reject $\mathcal{H}_0$ with a $p$-value $< 0.01$ and a large effect size (*i.e.,* more than 0.5) for all datasets, indicating that the observed difference in the memorization scores of $S_C$ and $S_E$ is statistically and practically significant. We also apply the same t-test for all

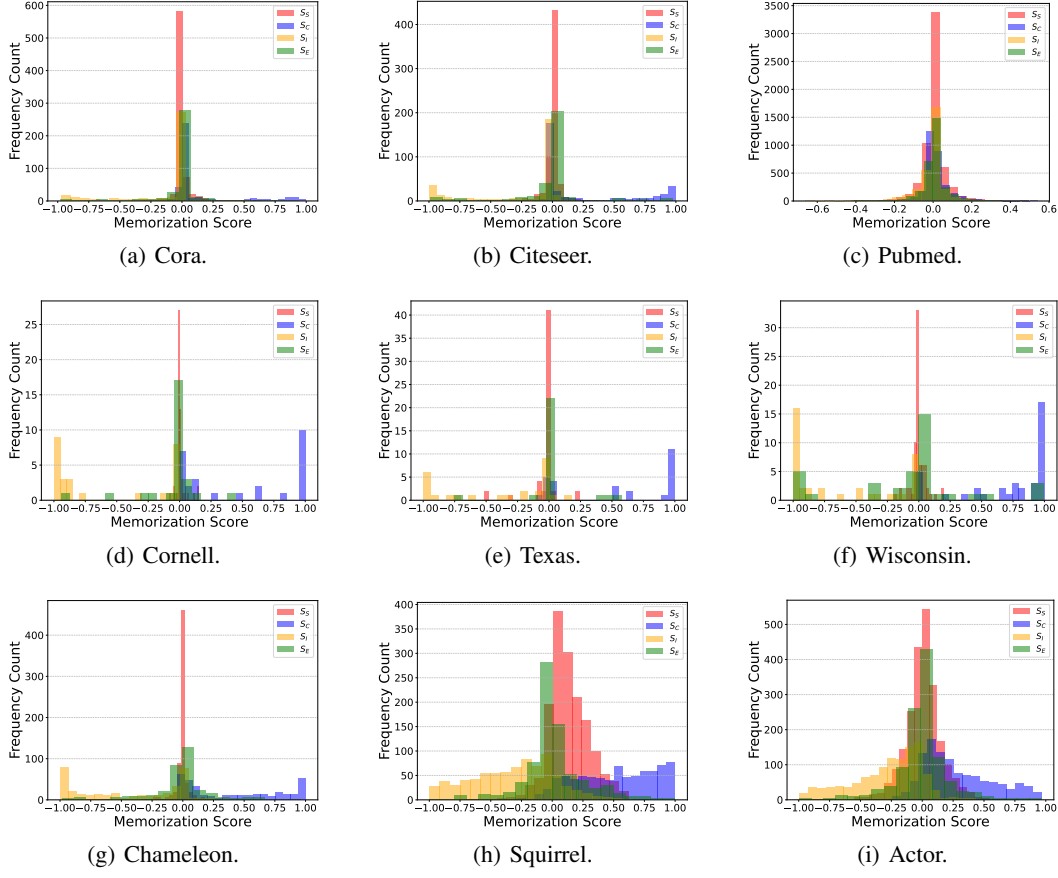

Figure 12: Distribution of memorization scores for each node category $S_S$, $S_C$, $S_I$, and $S_E$, on 9 real-world datasets, trained on the GCN model.

data subsets. The results confirm that $S_C$ ($S_I$) is significantly more (less) memorized by the GNN model than $S_S$ and $S_E$.

### E.3 Larger Datasets

The leave-one-out memorization [21] is theoretically well-founded and provides a precise notion of per-sample-level memorization. While the ideal form requires training multiple models, in our work, we approximate it efficiently: Our method only requires a single additional trained model and evaluates the effect for simultaneous removal of batches of nodes (25% of training nodes). This keeps the overhead minimal while still aligning closely with the theoretical definition. We present additional results on larger datasets such as Photo, Physics, Computer [57] and ogb-arxiv [30] datasets in Table 7. We can observe that our proposed framework scales well to large graphs and the datasets exhibit low memorization rates as they are highly homophilic which aligns with our Proposition 1.

### E.4 Graph Transformer

We also experiment with a Graph Transformer (GT), we use the architecture proposed in [50] with following modifications: we use 2 attention heads and 1-layer GT (Input → Linear Projection → [Attention + FFN Block] → Output Linear). The results on four real-world datasets are presented in Table 8. A caveat of applying our framework to study memorization in Graph Transformers is that it is hard to analyze the non-trivial effects the attention mechanism can have on memorization. Further, since a Transformer architecture is more general purpose than a GNN, it is unclear if a transformer shows an implicit bias to leverage the graph structure and a lack of NTK formulation for Graph Transformers makes it difficult to explain the emergence of memorization.

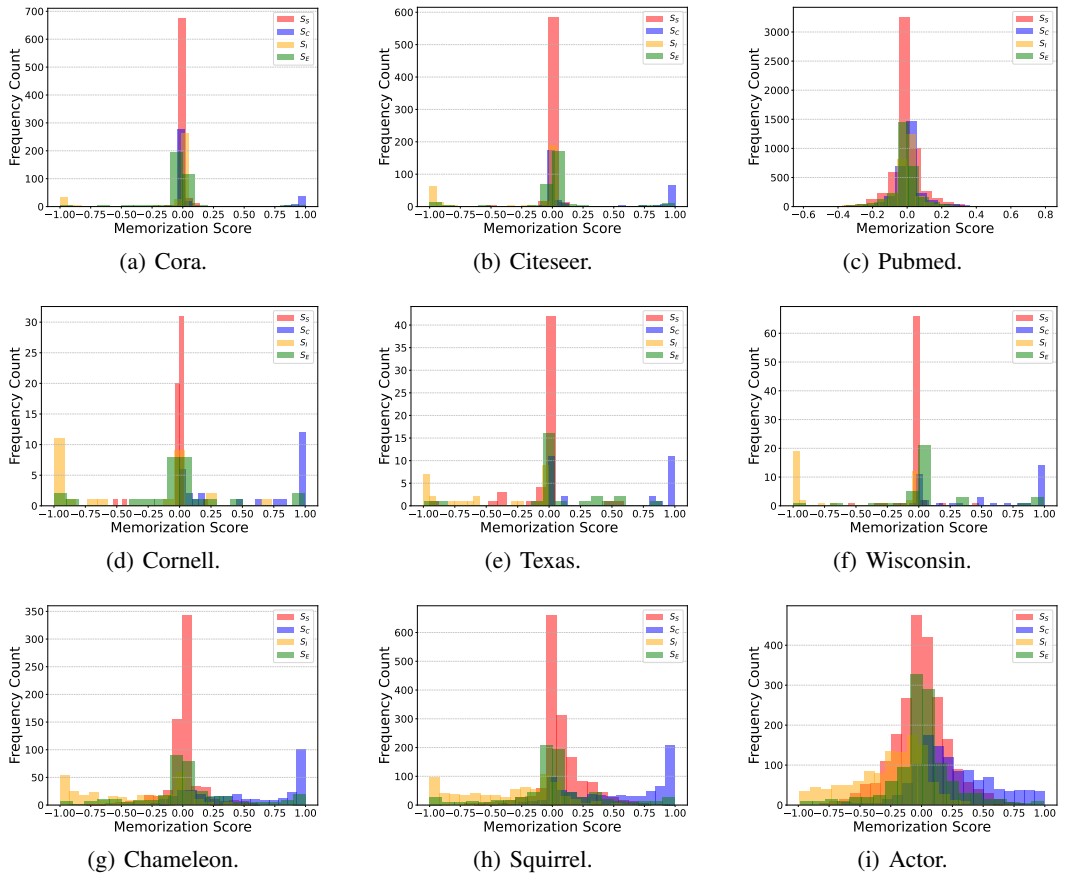

Figure 13: Distribution of memorization scores for each node category $S_S$, $S_C$, $S_I$, and $S_E$, on 9 real-world datasets, trained on the GATv2 model.

### E.5  2-hop aggregator methods

To analyze whether our findings on memorization are robust to the GNN's receptive field size, or if they are merely an artifact of a 1-hop message passing scheme, we conducted additional experiments with another GNN variant called Simplified Graph Convolution (SGC) [65], which is designed to analyze the impact of multi-hop neighborhoods by propagating features over a specified number of hops, $K$. We ran our experiments using SGC with $K = 2$, thereby allowing the model to aggregate information directly from the 2-hop neighborhood and present results on four real-world datasets in Table 9.

## F  Label Disagreement Score

**Choice of** $k$**.** An important hyperparameter for calculating the LDS is the value of $k$, in our case we set this value to $3$, here the idea is to mimic the $k$ hop aggregation step of the GNN model. By setting $k = 3$, we are trying to trace how the 3-hop neighborhood of the graph looks like for a GNN model that is trying to aggregate information, and if the label and feature distributions are uniform or do they have large *surprises* that can induce memorization. Setting a smaller $k$ value allows us to analyze the local neighborhood anomalies. It is possible to also calculate the LDS by setting higher $k$ values such as $5, 7, 10$, but the GNN models we use in practice do not have large receptive fields to aggregate information from higher order neighborhoods, unless they are explicitly designed to do so in cases like [47, 22, 20].

**Additional results.** In Table 10, we present the LDS for remaining datasets trained on GCN model. In Table 11 and Table 12 we present the LDS for datasets trained on GraphSAGE and

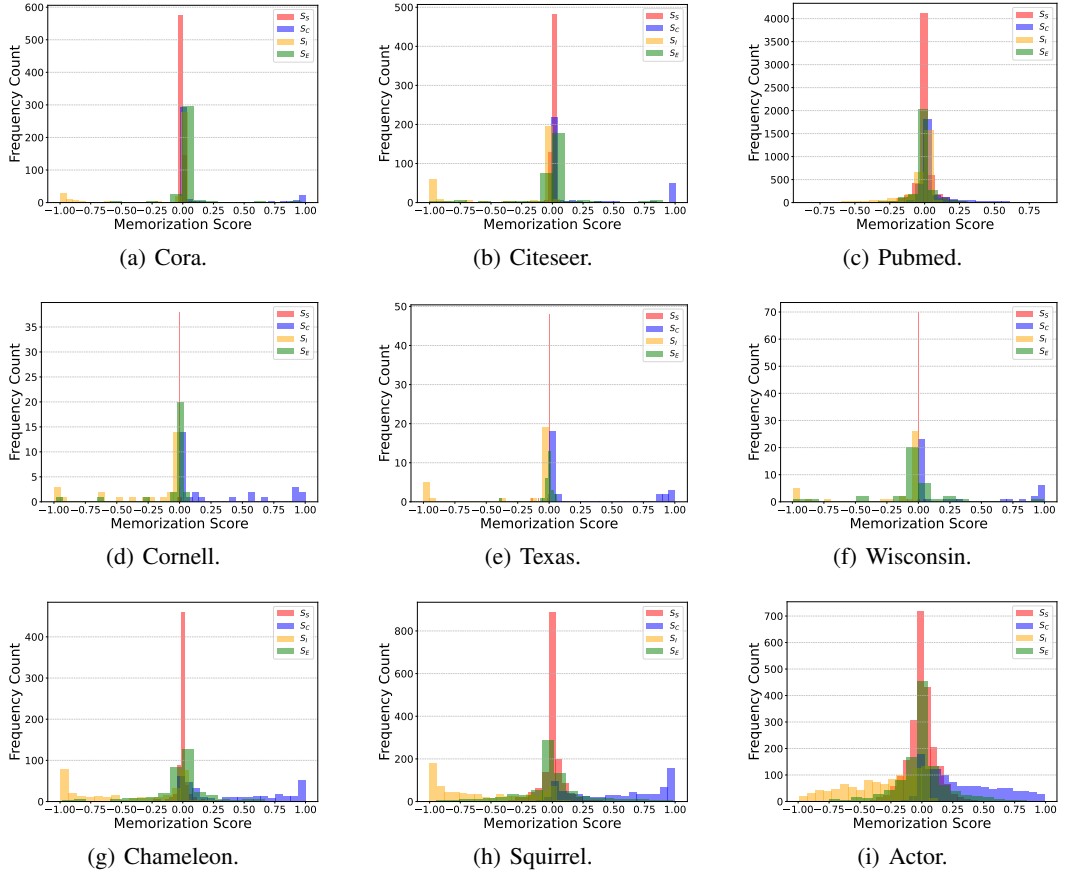

Figure 14: Distribution of memorization scores for each node category $S_S$, $S_C$, $S_I$, and $S_E$, on 9 real-world datasets, trained on the GraphSAGE model.

GATv2 respectively. Across these datasets, we can observe that our proposed metric consistently distinguishes memorized and non-memorized nodes through significantly higher LDS for memorized nodes than non-memorized ones. The metric also explains why certain datasets are potentially resistant to memorization. For instance, in Figure 12(c) and in Table 10, we can observe that Pubmed shows little to no candidate nodes as memorized, indicating both the models $f$ and $g$ output high confidence. A potential reason for such consistent performance of the models can be attributed to a uniform neighborhood, that is, there are no *surprise* nodes with unusual labels or features that might require memorization to learn.

**Nuances of LDS.** However, we also have to note certain shortcomings of our proposed metric, for instance the LDS for Actor dataset trained on GCN, GraphSAGE and GATv2 consistently yield statistically less significant scores to distinguish memorized and non-memorized nodes, we surmise this does not necessarily mean a failure of our metric, but rather the inability of the metric to capture the nuances of the dataset itself. From Table 18, we can observe that the Actor dataset is highly heterophilic and the node label informativeness is very low, thus the neighborhood of the dataset is highly non-uniform to yield consistent label disagreement scores. It is also possible the different aggregation methods used, for instance the attention mechanism in GATv2 or concatenating the final feature vectors instead of simple mean aggregation in GraphSAGE could also affect the LDS calculation. Further, in Table 10 and Table 11, we get $\infty$ for the effect sizes of Pubmed and, Wisconsin and Texas. For Pubmed, the reason is because there are either 1 or no nodes showing memorization and for datasets like Wisconsin and Texas, these are very small heterophilic datasets and the variance of scores for memorized and non-memorized is $0.0$, hinting they all have similar LDS values. While this highlights a technical limitation in applying standard variance-based effect size metrics in these edge cases, it primarily reflects the highly constrained nature of local neighborhoods and the resulting

Table 6: Results of statistical t-test for comparing memorization scores between different categories on 9 real-world datasets, trained on GCN. It is confirmed that $S_C$ ($S_I$) is significantly more (less) memorized by the model than $S_S$ and $S_E$.

|  | Cora | | Citeseer | | Pubmed | |
|---|---|---|---|---|---|---|
| Null Hypothesis | p-value | Effect size | p-value | Effect size | p-value | Effect size |
| $\mathcal{M}(S_C) \leq \mathcal{M}(S_S)$ | 0.000 | 0.5421 | 0.000 | 0.9483 | 0.000 | 0.2427 |
| $\mathcal{M}(S_C) \leq \mathcal{M}(S_E)$ | 0.000 | 0.4211 | 0.000 | 0.7998 | 0.000 | 0.2365 |
| $\mathcal{M}(S_S) \leq \mathcal{M}(S_I)$ | 0.000 | 0.7227 | 0.000 | 0.8617 | 0.000 | 0.2103 |
| $\mathcal{M}(S_E) \leq \mathcal{M}(S_I)$ | 0.000 | 0.5625 | 0.000 | 0.6442 | 0.000 | 0.1989 |

|  | Cornell | | Texas | | Wisconsin | |
|---|---|---|---|---|---|---|
| Null Hypothesis | p-value | Effect size | p-value | Effect size | p-value | Effect size |
| $\mathcal{M}(S_C) \leq \mathcal{M}(S_S)$ | 0.002576 | 0.9163 | 0.000 | 1.4675 | 0.000 | 1.4941 |
| $\mathcal{M}(S_C) \leq \mathcal{M}(S_E)$ | 0.000831 | 0.9849 | 0.000 | 1.3502 | 0.000 | 1.1926 |
| $\mathcal{M}(S_S) \leq \mathcal{M}(S_I)$ | 0.000002 | 1.6713 | 0.000 | 1.6077 | 0.000 | 2.1659 |
| $\mathcal{M}(S_E) \leq \mathcal{M}(S_I)$ | 0.000365 | 1.0621 | 0.000 | 1.3398 | 0.000 | 1.4669 |

|  | Chameleon | | Squirrel | | Actor | |
|---|---|---|---|---|---|---|
| Null Hypothesis | p-value | Effect size | p-value | Effect size | p-value | Effect size |
| $\mathcal{M}(S_C) \leq \mathcal{M}(S_S)$ | 0.000 | 1.6719 | 0.000 | 1.4719 | 0.000 | 1.2230 |
| $\mathcal{M}(S_C) \leq \mathcal{M}(S_E)$ | 0.000 | 1.4670 | 0.000 | 1.5799 | 0.000 | 1.0820 |
| $\mathcal{M}(S_S) \leq \mathcal{M}(S_I)$ | 0.000 | 1.7081 | 0.000 | 1.5538 | 0.000 | 1.2615 |
| $\mathcal{M}(S_E) \leq \mathcal{M}(S_I)$ | 0.000 | 1.3474 | 0.000 | 1.1429 | 0.000 | 1.1544 |

Table 7: Memorization score (MemScore) and Memorization rate (MR) for the candidate nodes on four larger datasets, averaged over 3 random seeds, along with empirical runtimes.

| Datasets | Avg MemScore | MR (%) | Nodes | Edges | Homophily | Runtime (Seconds) |
|---|---|---|---|---|---|---|
| Photo | 0.0093±0.0084 | 0.1±0.0 | 7,650 | 238,162 | 0.7850 | 3.97±0.24 |
| Physics | 0.0156±0.0018 | 0.4±0.1 | 34,493 | 495,924 | 0.8724 | 17.82±0.24 |
| Computer | 0.0005±0.0234 | 0.6±0.9 | 13,752 | 491,722 | 0.6823 | 6.26±0.19 |
| ogb-arxiv | 0.0031±0.0051 | 0.4±0.3 | 169,343 | 1,166,243 | >0.5 | 491.01±1.55 |

LDS values on these specific small graphs, rather than necessarily invalidating the underlying concept of label disagreement. It indicates that on such graphs, the local disagreement patterns measured by LDS might be very uniform within certain node groups.

**Computational complexity of LDS.** Calculating the Label Disagreement Score (LDS), $\text{LDS}_k(v_i|S_C)$, requires finding the k-nearest neighbor set $G_{v_i}^{k-\text{NN}}$ for each node $v_i \in S_C$. We use the $l_2$ distance on input features $x \in \mathbb{R}^d$, a naive implementation involves pairwise distance computations and selection. For each $v_i$, this takes roughly $\mathcal{O}(|S_C|d + |S_C|\log|S_C|)$ time. Repeating for all nodes in $S_C$ yields an overall complexity of approximately $\mathcal{O}(|S_C|^2(d + \log|S_C|))$.

Utilizing optimized libraries like `scikit-learn`, often employing spatial indexing (e.g., k-d trees) built on the features of nodes in $S_C$, can improve average-case performance. The index construction takes roughly $\mathcal{O}(|S_C|d\log|S_C|)$ time. Querying for $k$ neighbors for each $v_i \in S_C$ then has an average cost potentially closer to $\mathcal{O}(k\log|S_C|)$. The overall optimized average-case complexity becomes approximately $\mathcal{O}(|S_C|d\log|S_C| + |S_C|k\log|S_C|)$.

Since $|S_C|$ typically scales with the total dataset size $N$ (i.e., $|S_C| \propto N$), the computational cost exhibits a super-linear dependence on $N$. The optimized complexity scales approximately as $\mathcal{O}(Nd\log N + Nk\log N)$. This dependence explains the increased runtime observed for larger datasets. While optimized implementations enhance feasibility, the necessity of indexing or querying within a node set whose size grows with $N$ fundamentally governs the scaling of runtime. We report the empirical runtime for calculating the LDS on all datasets trained on GCN, GraphSAGE and GATv2 in Table 19, Table 20 and Table 21. We can observe that the runtime for LDS computation is larger for larger graphs such as Pubmed, Squirrel and Actor datasets.

Table 8: Memorization score (MemScore) and Memorization rate (MR) for candidate nodes on four real-world datasets, averaged over 3 random seeds, with Graph Transformer as our backbone model.

| Datasets | Avg MemScore | MR (%) | Runtime (Seconds) |
|---|---|---|---|
| Cora | 0.1359±0.0072 | 13.4±0.7 | 1355.43±3.76 |
| Citeseer | 0.2486±0.0070 | 24.9±0.20 | 1158.74±7.11 |
| Chameleon | 0.2807±0.0150 | 27.3±1.3 | 1222.66±1.49 |
| Squirrel | 0.4810±0.0094 | 47.7±1.2 | 4748.68±41.52 |

Table 9: Memorization score (MemScore) and Memorization rate (MR) for candidate nodes on four real-world datasets, averaged over 3 random seeds, with SGC as our backbone model.

| Datasets | Avg MemScore | MR (%) |
|---|---|---|
| Cora | 0.0482±0.0165 | 3.8±0.4 |
| Citeseer | 0.1124±0.0213 | 9.7±2.8 |
| Chameleon | 0.3852±0.0111 | 37.3±0.8 |
| Squirrel | 0.3745±0.0141 | 34.5±3.2 |

# G   Strategies for Mitigating Memorization in GNNs

In Section 6 we discussed the practical implications of studying GNN memorization. We showed how memorization can put models at privacy risk as adversaries can easily infer sensitive training data. In this section, we propose a surprisingly simple yet effective strategy to mitigate memorization, namely *Graph Rewiring*, which is the process of modifying the edge structure of the graph based on some pre-defined criteria such as Ricci curvature [61, 25, 46], spectral gap [33, 32] or feature similarity [10, 54] and has been shown to mitigate issues like over-squashing [2] and over-smoothing [35] while also improving the generalization performance on downstream tasks. Our theoretical results in Section 4 demonstrate that one of the key factors influencing memorization in GNNs is the graph homophily. Ideally, we can optimize homophily directly to control the memorization. However, the challenge here is that calculating homophily requires access to all the node labels, which conflicts with the downstream task of predicting the labels. An alternative is to look for proxy criteria that can indirectly influence the homophily level.

**Graph Rewiring Reduces Memorization.** We adopt the graph rewiring framework proposed in [54], which adds or (and) deletes edges by computing the pair-wise cosine similarity. The edges are ranked and modified such that their inclusion/exclusion should lead to an increase in the mean feature similarity between the nodes. We defer the readers to [54] for more details on the rewiring framework. We hypothesize that graph rewiring based on feature similarity will indirectly improve the graph homophily, which should effectively decrease the memorization in GNNs. To test this hypothesis we train a GCN with 3 random seeds on the `syn-cora` dataset and compute memorization scores based on our framework (Section 3), we rewire the graph by adding or (and) deleting edges and recompute the memorization scores. We also measure the average reduction in memorization scores and memorization rate on the candidate nodes $S_C$ for the GNNs trained on rewired graphs compared to those trained on the original graphs. Note that, the number of edges to modify is a hyperparameter that needs to be tuned depending on the graph. We consider $\{100, 500, 1000\}$ as the space of edge budget for hyperparameter tuning. We choose the setting that involves minimal edge modifications while leading to maximum improvement in the model's test accuracy, maximum improvement in the graph homophily level, and maximum reduction in the memorization rate. We plot the average memorization rate of candidate nodes of `syn-cora-h0.0` before and after rewiring the graph for additions, both additions and deletions and deletions in Figure 15. We can observe that edge deletions has the most minimal impact on reducing memorization. We report the actual values for all the datasets in Table 14, Table 15 and Table 16.

**Edge Additions.** In Table 14, we present the results for memorization reduction when edges are added to the graph. We can observe, for instance, on `syn-cora-h0.0`, by adding edges, we improve the test accuracy by **8.85**% and reduce the memorization rate on candidate nodes by **5.11**%.

Table 10: Label disagreement score for memorized vs non-memorized nodes in the $S_C$ for remaining real-world datasets trained on GCN and averaged over 3 random seeds.

| Dataset | MemNodes LDS | Non-MemNodes LDS | p-value ($< 0.01$) | Effect Size |
|---|---|---|---|---|
| Pubmed | 0.33±0.00 | 0.29±0.00 | 0.00 | Inf |
| Cornell | 0.50±0.01 | 0.34±0.0085 | 0.006441 | 8.7681 |
| Texas | 0.73±0.0221 | 0.45±0.0254 | 0.010398 | 6.8804 |
| Wisconsin | 0.51±0.00 | 0.25±0.00 | 0.00 | Inf |
| Actor | 0.78±0.0032 | 0.76±0.0011 | 0.010259 | 6.9275 |

Table 11: Label disagreement score for memorized vs non-memorized nodes in the $S_C$ for 9 real-world datasets trained on GraphSAGE and averaged over 3 random seeds.

| Dataset | MemNodes LDS | Non-MemNodes LDS | p-value ($< 0.01$) | Effect Size |
|---|---|---|---|---|
| Cora | 0.7697±0.0283 | 0.6179±0.0043 | 0.017565 | 5.26 |
| Citeseer | 0.7850±0.0129 | 0.5863±0.0024 | 0.002069 | 15.52 |
| Pubmed | 0.6312±0.0160 | 0.2877±0.0007 | 0.000802 | 24.95 |
| Cornell | 0.80±0.0740 | 0.34±0.0128 | 0.009608 | 7.16 |
| Texas | 0.9048±0.00 | 0.3667±0.00 | 0.0 | inf |
| Wisconsin | 0.8467±0.0323 | 0.3520±0.0333 | 0.001785 | 16.71 |
| Chameleon | 0.7985±0.0023 | 0.7588±0.0017 | 0.003820 | 11.40 |
| Squirrel | 0.8390±0.0014 | 0.7814±0.0024 | 0.001727 | 16.99 |
| Actor | 0.7911±0.0054 | 0.7747±0.0021 | 0.073673 | 2.45 |

**Edge Deletions** In Table 15, we present the results for memorization reduction by deleting edges in the graph. We can observe that although we show memorization reduction in all the datasets except on dataset `syn-cora-h0.5`. By deleting edges, the memorization rate on `syn-cora-h0.5` increases by **2.55**% while test accuracy reduces by **0.87**%. This counter-result highlights the nuances of trying to mitigate memorization by graph rewiring. It is possible that there exists a different edge budget that could still improve the metrics on this dataset, which can only be found by tuning a large number of edge budgets and then measuring the memorization rate, which quickly becomes computationally expensive.

**Simultaneous Additions-Deletions.** To ensure we don't skew the edge distribution of the graph by adding or deleting edges, we also perform an experiment where we simultaneously add and delete edges to ensure the rewired graph doesn't deviate much from the original graph statistics. The results are presented in Table 16. We can observe that after simultaneously adding and deleting edges, the graph homophily level increases for all `syn-cora` graphs, leading to an decrease in the memorization score and memorization rate. At the same time, there is an improvement in the test accuracy of the models. For instance, on `syn-cora-h0.0`, by adding and deleting 1000 edges, we improve the test accuracy by **7.87**% and reduce the memorization rate on candidate nodes by **6.01**%.

### G.1 Graph Rewiring vs. Label Smoothing for Mitigating Memorization

An alternative approach to mitigate memorization is to add a regularization technique such as label smoothing which discourages overconfidence by training on soft probability distributions instead of hard one-hot labels. We compare label smoothing with $\epsilon = 0.1$ and graph rewiring in Table 17 on syn-cora datasets. Evidently, graph rewiring not only reduces memorization but also improves downstream performance, contrary to using label smoothing which has an inherent side-effect of also reducing the test accuracy along with the memorization rate. In the third column of Table 17, we combine both label smoothing and graph rewiring, to highlight that these methods are not mutually exclusive and can be combined, working both at the data level and the model level, to reduce memorization. Our results indicate that graph rewiring is by far the simplest and best strategy to reduce memorization without affecting the downstream performance.

Table 12: Label disagreement score for memorized vs non-memorized nodes in the $S_C$ for 9 real-world datasets trained on GATv2 and averaged over 3 random seeds.

| Dataset | MemNodes LDS | Non-MemNodes LDS | p-value $(< 0.01)$ | Effect Size |
|---|---|---|---|---|
| Cora | 0.7717±0.0102 | 0.6498±0.0009 | 0.003147 | 12.57 |
| Citeseer | 0.7276±0.0149 | 0.5723±0.0051 | 0.006398 | 8.79 |
| Pubmed | 0.4333±0.0815 | 0.2873±0.00 | 0.103304 | 2.02 |
| Cornell | 0.5534±0.0177 | 0.3967±0.0161 | 0.017247 | 5.31 |
| Texas | 0.6343±0.0122 | 0.3999±0.0125 | 0.003702 | 11.58 |
| Wisconsin | 0.4574±0.0595 | 0.3784±0.0628 | 0.409718 | 0.73 |
| Chameleon | 0.8868±0.0126 | 0.5934±0.0208 | 0.001779 | 16.74 |
| Squirrel | 0.8215±0.0093 | 0.7849±0.0101 | 0.094732 | 2.13 |
| Actor | 0.7920±0.0037 | 0.7715±0.0011 | 0.021174 | 4.78 |

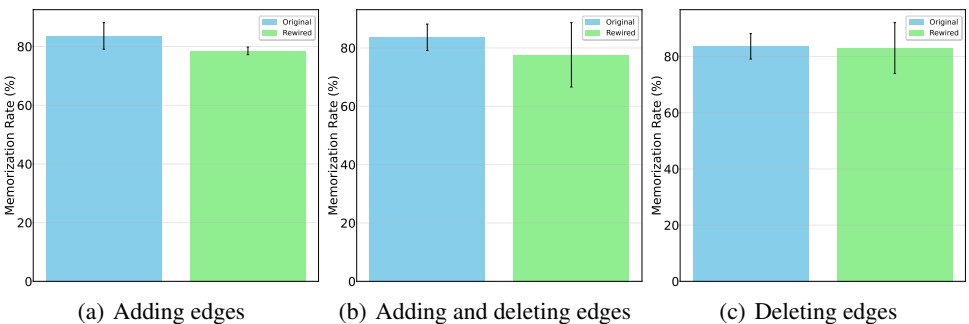

(a) Adding edges  (b) Adding and deleting edges  (c) Deleting edges

Figure 15: **Memorization rate of candidate nodes in `syn-cora-h0.0`, before and after edge modification.**

# H  Experimental setup and hyperparameters

## H.1  Graph Generation Process for `syn-cora`

We use the `syn-cora` dataset introduced in [68] to derive all our insights on memorization, as this benchmark provides the perfect setup to vary the graph homophily level and study its effects on the rate of memorization. The graph generation process follows a preferential-attachment setup [7] and is similar to the methods outlined in [34, 1]. The process involves starting with an initial graph and adding new nodes. We can connect a new node $u \in$ class $i$ to an existing node $v \in$ class $j$ based on some probability $p_{uv}$. To ensure the graphs generated follow a power law and the level of heterophily is controllable, a dependency on a class compatibility matrix $H_{ij} \in \mathbf{H}$ between class $i$ and $j$, and the degree $d_v$ of the existing node $v$ are introduced. The node features from real-world graphs, such as Cora [41], are sampled and assigned to the synthetically generated graphs. We use graphs with homophily levels $h = \{0.0, \ldots, 1.0\}$, where a score of $h = 0.0$ denotes a highly heterophilic graph and a homophily score $h = 1.0$ indicates a highly homophilic graph. We refer the readers to [68] for more details on how the compatibility matrix is defined and the graphs are generated. In our experiments, we do not generate the graphs ourselves.

## H.2  Hyperparameters

We use PyTorch Geometric [23] for all of our experiments. The hyperparameters used for training our models, the dataset statistics such as the number of nodes and edges, the homophily and label informativeness measures [49] are reported in Table 18. All of our experiments were done on a T4 GPU highlighting the computationally efficiency of our proposed framework.

Table 13: TPR at 1% FPR for MIA on GNNs trained on the original and rewired `syn-cora`.

| Dataset | TPR@1%FPR (Original) | TPR@1%FPR (Rewired) |
|---|---|---|
| `syn-cora-h0.00` | 0.0992 | 0.0090 |
| `syn-cora-h0.30` | 0.0455 | 0.0090 |
| `syn-cora-h0.50` | 0.0370 | 0.0135 |
| `syn-cora-h0.70` | 0.0380 | 0.0180 |
| `syn-cora-h1.00` | 0.0194 | 0.0195 |

Table 14: Memorization score (MemScore), Memorization rate (MR), Test accuracy and Homophily level for `syn-cora` dataset averaged over 3 random seeds, trained on GCN, before and after adding edges to the graph.

| Dataset | Avg MemScore Before | MR (%) Before | Test Accuracy Before | Homophily Before | Avg MemScore After | MR (%) After | Test Accuracy After | Homophily After | Edges Added |
|---|---|---|---|---|---|---|---|---|---|
| `syn-cora-h0.0` | 0.7091±0.0168 | 83.63±1.82 | 14.64±1.24 | 0.00 | 0.6563±0.0081 | 78.53±0.52 | 23.50±1.70 | 0.0159 | 1000 |
| `syn-cora-h0.3` | 0.5573±0.0047 | 63.96±1.19 | 33.33±0.94 | 0.3002 | 0.5226±0.0049 | 59.61±1.13 | 38.69±1.41 | 0.3659 | 1000 |
| `syn-cora-h0.5` | 0.4106±0.0126 | 45.95±1.62 | 63.28±4.31 | 0.5115 | 0.3992±0.0029 | 43.24±0.45 | 70.27±4.49 | 0.5477 | 1000 |
| `syn-cora-h0.7` | 0.2616±0.0070 | 25.98±1.45 | 77.27±2.86 | 0.6927 | 0.2457±0.0036 | 25.23±0.00 | 79.13±0.47 | 0.7020 | 500 |
| `syn-cora-h1.0` | 0.0021±0.0005 | 0.00 | 100 | 1.0 | 0.0018±0.0005 | 0.00 | 100 | 1.0 | 100 |

## H.3 Empirical runtimes

We present the empirical runtimes averaged over 3 random seeds in seconds for training models $f$, $g$ and calculating the memorization score for candidate nodes $S_C$ in Table 19, Table 20 and Table 21, where the backbone GNN models are GCN, GraphSAGE and GATv2 respectively. We can observe that our proposed label memorization framework `NCMemo` is consistently computationally friendly across datasets and different GNN models. We also report the runtime for calculating the label disagreement score in the same tables. We can see that our proposed framework for calculating the label disagreement score is computationally friendly for almost all the datasets and can be expensive for very large graphs.

Table 15: Memorization score (MemScore), Memorization rate (MR), Test accuracy and Homophily level for `syn-cora` dataset averaged over 3 random seeds, trained on GCN, before and after deleting edges to the graph.

| Dataset | Avg MemScore Before | MR (%) Before | Test Accuracy Before | Homophily Before | Avg MemScore After | MR (%) After | Test Accuracy After | Homophily After | Edges Deleted |
|---|---|---|---|---|---|---|---|---|---|
| syn-cora-h0.0 | 0.7091±0.0168 | 83.63±1.82 | 14.64±1.24 | 0.00 | 0.6994±0.0153 | 83.03±3.64 | 14.64±0.94 | 0.00 | 100 |
| syn-cora-h0.3 | 0.5573±0.0047 | 63.96±1.19 | 33.33±0.94 | 0.3002 | 0.5698±0.0082 | 63.51±1.19 | 34.43±2.15 | 0.3079 | 500 |
| syn-cora-h0.5 | 0.4106±0.0126 | 45.95±1.62 | 63.28±4.31 | 0.5115 | 0.4413±0.0026 | 48.50% ± 0.26 | 62.40±1.70 | 0.5117 | 100 |
| syn-cora-h0.7 | 0.2616±0.0070 | 25.98±1.45 | 77.27±2.86 | 0.6927 | 0.2492±0.0041 | 25.08±0.52 | 76.39±4.95 | 0.6945 | 100 |
| syn-cora-h1.0 | 0.0021±0.0005 | 0.00 | 100 | 1.0 | 0.0023±0.0005 | 0.00 | 100 | 1.0 | 100 |

Table 16: Memorization score (MemScore), Memorization rate (MR), Test accuracy and Homophily level for `syn-cora` dataset averaged over 3 random seeds, trained on GCN, before and after simultaneous edge additions and deletions.

| Dataset | Avg MemScore Before | MR (%) Before | Test Accuracy Before | Homophily Before | Avg MemScore After | MR (%) After | Test Accuracy After | Homophily After | Edges Modified |
|---|---|---|---|---|---|---|---|---|---|
| syn-cora-h0.0 | 0.7091±0.0168 | 83.63±1.82 | 14.64±1.24 | 0.00 | 0.6804±0.0288 | 77.63±4.44 | 22.51±0.47 | 0.0159 | 1000 |
| syn-cora-h0.3 | 0.5573±0.0047 | 63.96±1.19 | 33.33±0.94 | 0.3002 | 0.5210±0.0143 | 58.71±0.69 | 37.16±2.05 | 0.3501 | 500 |
| syn-cora-h0.5 | 0.4106±0.0126 | 45.95±1.62 | 63.28±4.31 | 0.5115 | 0.3871±0.0153 | 39.79±1.45 | 65.79±1.88 | 0.5665 | 1000 |
| syn-cora-h0.7 | 0.2616±0.0070 | 25.98±1.45 | 77.27±2.86 | 0.6927 | 0.2413±0.0058 | 24.32±1.56 | 78.91±2.49 | 0.7114 | 500 |
| syn-cora-h1.0 | 0.0021±0.0005 | 0.00 | 100 | 1.0 | 0.0022±0.0005 | 0.00 | 100 | 1.0 | 100 |

Table 17: Memorization score (MemScore), Memorization rate (MR) for candidate nodes on `syn-cora`, averaged over 3 random seeds, trained on GCN, with Graph Rewiring, Label Smoothing, and Graph Rewiring + Label Smoothing.

| Datasets | Graph Rewiring Avg MemScore | Graph Rewiring MR (%) | Graph Rewiring Test Accuracy | Label Smoothing Avg MemScore | Label Smoothing MR (%) | Label Smoothing Test Accuracy | Graph Rewiring + Label Smoothing Avg MemScore | Graph Rewiring + Label Smoothing MR (%) | Graph Rewiring + Label Smoothing Test Accuracy |
|---|---|---|---|---|---|---|---|---|---|
| syn-cora-h0.0 | 0.6563±0.0081 | 78.53±0.52 | 23.50±1.70 | 0.6000±0.0000 | 73.6±2.70 | 15.30±0.83 | 0.6050±0.0130 | 71.3±0.9 | 16.17±0.50 |
| syn-cora-h0.3 | 0.5226±0.0049 | 59.61±1.13 | 38.69±1.41 | 0.4836±0.0000 | 56.2±0.90 | 30.71±0.68 | 0.4665±0.0239 | 52.9±3.6 | 33.01±2.18 |
| syn-cora-h0.5 | 0.3992±0.0029 | 43.24±0.45 | 70.27±4.49 | 0.3728±0.0100 | 39.2±1.20 | 54.54±0.68 | 0.3640±0.0167 | 39.0±3.2 | 57.81±1.00 |
| syn-cora-h0.7 | 0.2457±0.0036 | 25.23±0.00 | 79.13±0.47 | 0.2667±0.0057 | 24.5±0.9 | 73.77±0.33 | 0.2506±0.0158 | 20.0±2.9 | 73.77±0.57 |
| syn-cora-h1.0 | 0.0018±0.0005 | 0.00 | 100 | 0.0265±0.0016 | 0 | 100 | 0.0282±0.0027 | 0 | 100 |

Table 18: Hyperparameters and dataset statistics.

| Dataset | Hidden Dimension | LR | #Layers | $|\mathcal{V}|$ | $|\mathcal{E}|$ | # Classes | Homophily Level | Node Label Informativeness |
|---|---|---|---|---|---|---|---|---|
| Cora | 32 | 0.01 | 3 | 2708 | 10138 | 7 | 0.7637 | 0.5763 |
| Citeseer | 32 | 0.01 | 3 | 3327 | 7358 | 6 | 0.6620 | 0.4653 |
| Pubmed | 32 | 0.01 | 3 | 19717 | 88648 | 3 | 0.6860 | 0.4223 |
| Cornell | 128 | 0.001 | 3 | 183 | 298 | 5 | -0.2029 | 0.1574 |
| Texas | 128 | 0.001 | 3 | 183 | 325 | 5 | -0.2260 | 0.0186 |
| Wisconsin | 128 | 0.001 | 3 | 251 | 515 | 5 | -0.1323 | 0.0874 |
| Chameleon | 128 | 0.001 | 3 | 2277 | 36101 | 5 | 0.0339 | 0.0516 |
| Squirrel | 128 | 0.001 | 3 | 5201 | 217073 | 5 | 0.0074 | 0.0277 |
| Actor | 128 | 0.001 | 3 | 7600 | 30019 | 5 | 0.0062 | 0.0018 |
| syn-cora | 32 | 0.01 | 3 | 1490 | 2965 to 2968 | 5 | 0.0 to 1.0 | 0.0 to 1.0 |

Table 19: Empirical runtime for training GCN models $f, g$ and calculating the memorization scores for $S_C$ and calculating the LD score averaged over 3 random seeds with 95% CI (confidence interval).

| Dataset | MemorizationScore Runtime (in seconds) | LDS Runtime (in seconds) |
|---|---|---|
| Cora | 2.16±0.52 | 7.31±0.16 |
| Citeseer | 1.93±0.15 | 10.52±0.28 |
| Pubmed | 2.86±0.15 | 41.63±0.72 |
| Cornell | 2.11±0.23 | 3.06±0.51 |
| Texas | 1.83±0.16 | 3.88±0.41 |
| Wisconsin | 2.00±0.44 | 3.50±0.07 |
| Chameleon | 2.36±0.17 | 8.27±0.45 |
| Squirrel | 6.23±0.21 | 24.49±0.11 |
| Actor | 2.92±0.20 | 25.14±0.28 |

Table 20: Empirical runtimes for training GraphSAGE models $f, g$ and calculating the memorization scores for $S_C$ and calculating the LD score averaged over 3 random seeds with 95% CI (confidence interval).

| Dataset | MemorizationScore Runtime (in seconds) | LDS Runtime (in seconds) |
|---|---|---|
| Cora | 1.70±0.19 | 7.24±0.45 |
| Citeseer | 2.55±0.15 | 10.27±0.26 |
| Pubmed | 4.30±0.23 | 69.71±0.30 |
| Cornell | 1.40±0.24 | 3.04±0.26 |
| Texas | 1.55±0.40 | 3.74±0.34 |
| Wisconsin | 1.39±0.15 | 3.10±0.56 |
| Chameleon | 5.29±0.16 | 8.35±0.46 |
| Squirrel | 24.51±0.13 | 25.56±0.16 |
| Actor | 3.83±0.23 | 24.25±0.53 |

Table 21: Empirical runtimes for training GAT models $f, g$ and calculating the memorization scores for $S_C$ and calculating the LD score averaged over 3 random seeds with 95% CI (confidence interval).

| Dataset | MemorizationScore Runtime (in seconds) | LDS Runtime (in seconds) |
|---|---|---|
| Cora | 3.24±0.26 | 8.38±0.45 |
| Citeseer | 3.52±0.18 | 10.60±0.27 |
| Pubmed | 11.97±0.17 | 70.26±0.81 |
| Cornell | 2.99±0.25 | 3.17±0.35 |
| Texas | 3.23±0.30 | 3.55±1.03 |
| Wisconsin | 2.96±0.18 | 3.22±0.45 |
| Chameleon | 13.75±0.16 | 8.18±0.36 |
| Squirrel | 64.81±0.48 | 26.60±0.27 |
| Actor | 17.47±0.21 | 25.57±0.25 |

