# OpenReview forum: "Memorization in Graph Neural Networks"
_NeurIPS.cc/2025/Conference — NeurIPS 2025 poster_

### Official Review · Reviewer_RAPS · 2025-06-23

**Clarity:** 3
**Significance:** 3
**Originality:** 4
**Rating:** 5
**Confidence:** 3

**Summary:**

The paper introduces NCMemo, a novel framework for measuring per-node label memorization in semi-supervised graph-node classification. This work provides a deep investigation into why and how GNNs memorize, links memorization to graph properties such as homophily, and proposes a practical method for mitigating resulting privacy vulnerabilities through graph rewiring.

**Questions:**

Q1. Could the edge additions/deletions that reduce memorization negatively impact tasks where edge semantics are important (e.g., recommender systems)? Some discussion or a case study would strengthen the practical claims.

**Ethical Concerns:**

["NO or VERY MINOR ethics concerns only"]

**Final Justification:**

My question is basically addressed. I believe this paper provides interesting insights into the fundamental properties of GNN models.

**Limitations:**

yes

**Quality:**

3

**Strengths And Weaknesses:**

**Strengths**

- The paper presents interesting insights by extending leave-one-out memorization analysis to the graph domain, and it is the first to attribute memorization to individual nodes.
- This work builds on insightful training dynamics analysis, incorporating NTK alignment curves to convincingly link increased memorization to the model’s structural bias in cases where kernel-target alignment is poor.
- The experimental results span three GNN backbones, multiple random seeds, statistical significance testing, and ablation studies (edge addition, deletion, and their combination), lending robustness to the conclusions.

**Weaknesses**

- Rewiring is not compared against other defenses (e.g., label smoothing), and its impact on downstream tasks that rely on the original edge topology is not evaluated.
- The memorization indicator uses a fixed threshold of τ = 0.5 (line 122) without analysis, which may make the conclusions threshold-dependent.

---

> ### Author Rebuttal · Authors · 2025-07-30
>
> We thank the Reviewer for recognizing NCMemo as a novel framework for measuring per-node memorization in GNNs. We appreciate their acknowledgment of our deep investigation into the mechanisms behind GNN memorization, the connection to graph homophily, and the practical privacy mitigation approach through graph rewiring. Below we address the comments and questions.
>
> >**Comparing rewiring against other defenses, e.g. label smoothing**
>
> Following the Reviewer’s suggestion, we include additional baselines. Eventually, we experiment with three approaches:
>
> 1. **Graph Rewiring** (Method described in the original submission): This technique relies purely on topological information such as node features indirectly improving the graph homophily level.
> 2. **Label Smoothing**: This regularization technique discourages overconfidence by training on soft probability distributions instead of hard one-hot labels. We use $\epsilon = 0.1$ for our experiments.
> 3. **Rewiring+ Label smoothing**: To analyze their interaction, we combined both methods.
>
> `Graph Rewiring:`
>
> |Datasets|AvgMemScore|MR(%)|TestAccuracy|
> |---|---|---|---|
> |syn-cora-h0.0|0.6563±0.0081|78.53±0.52|23.50±1.70|
> |syn-cora-h0.3|0.5226±0.0049|59.61±1.13|38.69±1.41|
> |syn-cora-h0.5|0.3992±0.0029|43.24±0.45|70.27±4.49|
> |syn-cora-h0.7|0.2457±0.0036|25.23±0.00|79.13±0.47|
> |syn-cora-h1.0|0.0018±0.0005|0.00|100|
>
> `Label Smoothing:`
>
> |Datasets|AvgMemScore|MR(%)|TestAccuracy|
> |---|---|---|---|
> |syn-cora-h0.0|0.6000±0.0000|73.6±2.70|15.30±0.83|
> |syn-cora-h0.3|0.4836±0.0000|56.2±0.90|30.71±0.68|
> |syn-cora-h0.5|0.3728±0.0100|39.2±1.20|54.54±0.68|
> |syn-cora-h0.7|0.2667±0.0057|24.5±0.9|73.77±0.33|
> |syn-cora-h1.0|0.0265±0.0016|0|100|
>
> `Rewiring+Label Smoothing:`
>
> |Datasets|AvgMemScore|MR(%)|TestAccuracy|
> |---|---|---|---|
> |syn-cora-h0.0|0.6050±0.0130|71.3±0.9|16.17±0.50|
> |syn-cora-h0.3|0.4665±0.0239|52.9±3.6|33.01±2.18|
> |syn-cora-h0.5|0.3640±0.0167|39.0±3.2|57.81±1.00|
> |syn-cora-h0.7|0.2506±0.0158|20.0±2.9|73.77±0.57|
> |syn-cora-h1.0|0.0282±0.0027|0|100|
>
> Our results show that *Label Smoothing* and *Rewiring+Label Smoothing*, while lowering the memorization rate, they also generally reduce the test accuracy, compared to the original performance before applying defenses in Table 10, Appendix G. In contrast, graph rewiring improves the downstream performance, i.e., demonstrating a clear performance trade-off.
>
> Finally, to show that this finding is independent of the choice of $\epsilon$ in Label Smoothing, we also include an ablation study with different $\epsilon = 0.1,0.3,0.5,0.7,0.9$ on syn-cora-h0.0. The results suggest that under each $\epsilon$, Label Smoothing yields a lower performance than rewiring, confirming this inherent trade-off.
>
> `syn-cora-h0.0:`
>
> | $\epsilon$ | AvgMem Score     | MR (%)         | Test Accuracy   |
> |------------|------------------|----------------|-----------------|
> | 0.0        | 0.6986 ± 0.01     | 82.4 ± 1.4     | 14.86 ± 1.37    |
> | 0.1        | 0.6000 ± 0.00     | 73.6 ± 2.70    | 15.30 ± 0.85    |
> | 0.3        | 0.4348 ± 0.0028   | 37.7 ± 1.4     | 15.19 ± 0.68    |
> | 0.5        | 0.2877 ± 0.0148   | 4.4 ± 3.2      | 14.75 ± 0.66    |
> | 0.7        | 0.1523 ± 0.0015   | 0              | 14.32 ± 0.38    |
> | 0.9        | 0.0188 ± 0.0036   | 0              | 18.80 ± 2.98    |
>
> >**Evaluating rewiring’s impact on downstream tasks that rely on original topology**
>
> To directly evaluate the rewiring’s impact on downstream tasks, we presented the testing accuracy before and after applying rewiring on synthetic datasets in Tables 10-12 in Appendix G. Additionally, we also present the downstream task performance before and after applying rewiring on four real-world graphs, in the following table. For simultaneous additions and deletions, the edge budget is 100. For only additions and deletions, the edge budgets explored are $\{5,10,50,100,500,1000,1500\}$ and the best performance is reported.
>
> `Comparing test accuracy before and after rewiring:`
> |Datasets|TestAccBefore|TestAccAfter (Adding edges)|TestAccAfter (Deleting edges)|TestAccAfter (Add+Del)|
> |---|---|---|---|---|
> |Cora|85.24±0.51|87.33±0.30|87.32±0.31|86.90±0.30|
> |Citeseer|77.52±1.63|78.07±0.40|79.01±0.38|78.57±0.36|
> |Chameleon|33.92±0.80|40.48±0.68|39.58±0.65|40.07±0.69|
> |Squirrel|26.79±0.76|34.64±0.36|34.07±0.43|33.56±0.43|
>
> In line with graph rewiring approaches [57,32,25,31,51], the downstream performance improves over baselines highlighting the fact that the graph with original topology is usually suboptimal for the task, while rewiring improves.
>
> >**Could the edge additions/deletions that reduce memorization negatively impact tasks where edge semantics are important (e.g., recommender systems)? Some discussion or a case study would strengthen the practical claims.**
>
> Graph rewiring is a well-established pre-processing technique from prior work that we adopt [57,32,25,31,51]. It is motivated by the common problem that real-world graph topologies are often suboptimal for GNNs due to bottlenecks etc.
>
> As demonstrated in our paper (Appendix G, Tables 10,11,12), our application of feature similarity based rewiring [51] indirectly influences the graph homophily and is highly effective in **reducing memorization while simultaneously improving downstream performance** on node classification. This confirms its utility as a practical tool for mitigating the specific memorization issues we identify.
>
> We agree that for tasks where edges have strict semantic meaning (e.g., chemical bonds in molecular graphs), modifying the graph without domain knowledge could be detrimental. Understanding how to perform "semantically-aware" rewiring in such domains is an important but orthogonal research question that is beyond the scope of our study.
>
> >**Memorization indicator uses a fixed threshold of $\tau = 0.5$**
>
> We chose the threshold provided in the well-established foundational work on memorization by Feldman [21]. This allows our results to be directly comparable and situated within the broader literature on this topic. It is important to note that our main results (e.g., in Figure 2 of the paper) present the full, un-thresholded distribution of memorization scores to provide a complete picture; the threshold is only used for calculating aggregate rates like the Memorization Rate (MR).
>
> To address the Reviewer’s comment, we also conducted an additional threshold-sensitivity analysis, reporting the average memorization score and memorization rate over various thresholds for the synthetic datasets and 4 real-world graphs Cora, Citeseer, Chameleon and Squirrel using GCN.
>
>
> `Synthetic Datasets:`
>
> ||**$\tau > 0.4$**||**$\tau > 0.6$**||**$\tau > 0.8$**||**$\tau > 0.9$**||
> |---|---|---|---|---|---|---|---|---|
> |Datasets|AvgMemScore|MR(%)|AvgMemScore|MR(%)|AvgMemScore|MR(%)|AvgMemScore|MR(%)|
> |syn-cora-h0.0|0.7087±0.0069|90.09±0.45|0.6798±0.0221|70.72±5.19|0.7292±0.0171|53.30±1.45|0.7239±0.0356|23.87±5.08|
> |syn-cora-h0.3|0.5900±0.0134|70.87±0.26|0.5830±0.0197|60.96±2.48|0.6438±0.0093|48.95±2.27|0.5530±0.0270|22.82±1.82|
> |syn-cora-h0.5|0.4883±0.0098|54.35±0.26|0.5012±0.0138|52.25±1.19|0.4379±0.0226|30.18±3.15|0.4766±0.0305|21.02±2.31|
> |syn-cora-h0.7|0.2295±0.0121|26.28±2.64|0.2174±0.0149|19.22±0.94|0.2393±0.0032|15.77±0.45|0.2103±0.0027|7.96±0.52|
> |syn-cora-h1.0|0.0025±0.0008|0|0.0024±0.0005|0|0.0023±0.0004|0|0.0029±0.0008|0|
>
> `Real-world graphs:`
>
> | |**$\tau > 0.4$**||**$\tau > 0.6$**||**$\tau > 0.8$**||**$\tau > 0.9$**||
> |---|---|---|---|---|---|---|---|---|
> |Datasets|AvgMemScore|MR(%)|AvgMemScore|MR(%)|AvgMemScore|MR(%)|AvgMemScore|MR(%)|
> |Cora|0.1340±0.0107|16.3±1.0|0.1128±0.0000|10.0±0.4|0.1311±0.0027|8.7±0.8|0.1004±0.0020|2.3±0.8|
> |Citeseer|0.2566±0.0118|30.0±1.8|0.2341±0.0055|23.0±2.3|0.2586±0.0100|18.0±4.0|0.2434±0.0114|12.4±2.0|
> |Chameleon|0.5297±0.0164|60.1±1.5|0.4898±0.0100|44.6±1.5|0.5391±0.0194|37.6±1.5|0.5549±0.0069|31.8±1.1|
> |Squirrel|0.4826±0.0275|57.0±2.7|0.4669±0.0100|37.9±3.1|0.4840±0.0187|22.6±2.5|0.4909±0.0258|11.5±1.0|
>
>
> Across all thresholds, we can see the same trend, i.e., memorization consistently decreases as graph homophily increases, confirming that our findings are robust to the selection of the threshold.

---

> > ### Comment · Reviewer_RAPS · 2025-08-05
> >
> > Thank you to the authors for addressing my question. I believe this paper provides interesting insights into the fundamental properties of GNN models. I will raise my score to Acceptance.

---

> > > ### Author Response · Authors · 2025-08-06
> > > **Thank you to the reviewer for engaging in the rebuttal**
> > >
> > > We kindly thank the reviewer for engaging with our rebuttal and for raising their score to Acceptance.

---

### Official Review · Reviewer_UeH1 · 2025-07-01

**Clarity:** 3
**Significance:** 3
**Originality:** 2
**Rating:** 2
**Confidence:** 4

**Summary:**

This paper presents the NCMemo, a framework to quantify label memorization in GNNs for semi-supervised node classification. Firstly,  author states the role of memorization in improving the performance of DNNs, and verifies the four proposed propositions on GNNs through many experimental results: (1) Higher graph homophily leads to lower memorization rates; (2) in low-homogeneity graphs, the optimization dynamics of GNNs show an over-reliance on the graph structure, leading to increased memorization rate; (3) better alignment between the kernel and the target or graph structure reduces memorization; (4) nodes with high label inconsistency in their feature space neighborhood are more prone to memorization.

**Questions:**

1.  The authors pointed out that due to the non-IID properties in graphs, analyzing the model’s ability to memorize individual nodes is challenging, as it requires modeling a complex interplay of factors. So, is it appropriate to define the GNN's memorization based on a single node (i.e., leave-one-out)?

2. In addition, I suggest adding research about the 2-hop topology on graphs, which is also important for the message passing mechanism of GNNs.

3. Would it be better to fit the results reflected in Fig. 5 and 7 using a quadratic polynomial?

4. Others please refer to the Weaknesses part.

**Ethical Concerns:**

["NO or VERY MINOR ethics concerns only"]

**Final Justification:**

After reading the paper again, I think this paper has major flaws, which are explained in my response to the authors' rebuttal. Thus, I have decided to decrease the score.

**Limitations:**

Yes

**Paper Formatting Concerns:**

None.

**Quality:**

2

**Strengths And Weaknesses:**

Strengths:

1. The paper migrates the concept of memorization from DNN to the GNN field and put forward some interesting propositions and discoveries, which fill the gap that similar studies on data memorization in GNNs are lacking.

2. The experiments in the paper are sufficient, and satisfactory experimental results have been obtained on both synthetic datasets and real-world datasets.

Weaknesses:

1. Some descriptions in the paper are not clear, such as the method to divide the training set into three disjoint partitions and Eqn. (1).

2. Lack of solid theoretical proof and derivation.

3. The position of Fig. 3 is not suitable.

4. The paper directly borrows a lot of methods and ideas from other papers, resulting in low innovation of this paper.

---

> ### Author Rebuttal · Authors · 2025-07-30
>
> We thank the Reviewer for their thorough assessment and for acknowledging that our work fills an important gap by studying memorization in GNNs and by highlighting that our empirical evaluation is *sufficient and satisfactory*. In the following, we address the Reviewer’s comments individually.
>
> >**Describing the method on data partitioning**
>
> We describe the data partitioning in Section 5 (Experimental setup) of our paper. To further clarify, we follow the setup proposed in [59] and divide the training data into three disjoint subsets. We divide our datasets into 60%/20%/20% as training, testing, and validation sets, respectively. Out of the 60% training set S, we further randomly divide the datasets into three disjoint subsets with ratios $S_S$ = 50%, $S_C$ = 25%, $S_I$ = 25%. We use these sets to train two models. Model $f$ is our target model whose memorization we want to evaluate. Model $g$ is an independent model that we use to calibrate according to Equation (1), following the leave-one-out-style definition of memorization by Feldman [21]. Model $f$ is trained on $S_S \cup S_C$, and model $g$ is trained on $S_S \cup S_I$. $S_C$ is the candidate set, i.e., the nodes whose memorization we want to quantify. Models $f$ and $g$ are both trained on $S_S$, but only model $f$ is trained on $S_C$. Therefore, the difference in behavior between model $f$ and $g$ on $S_C$ results from the fact that $f$ has seen the data and $g$ has not, allowing us to quantify the memorization. Including $S_I$ into the training set of $g$ makes $f$ and $g$ have the same number of training data points, which allows model $g$ to reach the performance of model $f$. This makes sure that differences in behavior are not due to dataset differences. We added an extended section with this explanation to Appendix H.
>
> >**Describing Equation (1)**
>
> Equation (1) follows the standard leave-one-out-style definition of memorization [21]. It operates on two models ($f$ and $g$) trained on the same dataset $S$, with the difference that for $f$, $S$ includes node $v_i$ and for $g$ it does not. The node $v_i$ is then the one whose memorization we aim to measure. The memorization is measured over the expectation of outcomes of the training algorithm. This means the expectation of behavior of models $f$ and $g$ on predicting the label of $v_i$ correctly as $y_i$. Intuitively, if $f$ is, on expectation, more capable of correctly predicting the label than $g$, this must result from the sole difference between the models, namely that $f$ has seen $v_i$ and $g$ has not. It hence quantifies $f$’s memorization on $v_i$. We also extended the explanation in Appendix H
>
> >**Lack of proofs**
>
> We believe our paper's primary contribution is a comprehensive **empirical investigation** into the **under-explored phenomenon of memorization in GNNs**. Our work provides the first formal framework to address this problem, supported by rigorous experiments across 9 real-world datasets and 5 synthetic datasets with 3 GNN architectures, extensive analysis of training dynamics, and a new node-level metric (LDS). We extended our work with a more formal proof of Proposition 2, added to the appendix. Here, we provide the sketch of the proof.
>
> `Proof sketch:`
> We delineate two regimes of learning:
>
> 1. **High-homophily settings:** In high homophily settings the GNN's inductive bias aligns with an informative graph structure, leading to a low generalization gap. Feldman's [21] Lemma 4.2 relates the generalization gap to the average memorization score. Given some distribution $P$, the learning algorithm $\mathcal{A}$, dataset $\mathcal{S}$, $\mathcal{E}^{train}$ is training error and $\mathcal{E}^{gen}$ test error we have
>
> $\frac{1}{n}\mathbb{E}_{\mathcal{S} \sim P^n} \left[\sum^{i \in [n]} mem (\mathcal{A},\mathcal{S}, i) \right] = \mathbb{E}[\mathcal{E}^{train}] - \mathbb{E}[\mathcal{E}^{gen}] \approx 0$. (High homophily $\implies$ low memorization)
>
> 2. **Low-homophily settings:** For an atypical node $v_i$ with high LDS, the model trained without it, $f_{\mathcal{S} \setminus \{v_i\}}$, relies on uninformative graph, resulting in a high leave-one-out error $\mathbb{P}_ {f \sim \mathcal{A}(\mathcal{S} \setminus \{v_i\})}[f(v_i) \neq y_i] \to 1$. To achieve 0 training error $\mathbb{P}_ {f \sim \mathcal{A}(\mathcal{S})}[f(v_i) \neq y_i] = 0$, the model needs to memorize.
>
> Feldman's Lemma 4.3 states
> $\mathbb{P}_ {f \sim \mathcal{A}(\mathcal{S})}[f(v_i) \neq y_i]  = \mathbb{P}_{f \sim \mathcal{A}(\mathcal{S} \setminus \{v_i\})}[f(v_i) \neq y_i] - mem(\mathcal{A}, \mathcal{S}, i)$.
>
> By substitution, we get $0 \approx 1 - {mem}(\mathcal{A}, \mathcal{S}, i) \implies {mem}(\mathcal{A}, \mathcal{S}, i) \approx 1$. Thus, in low-homophily, the model must force a high memorization to fit atypical nodes, confirming our empirical findings.
>
> >**Relying on methods from other papers and low innovativeness**
>
> We respectfully disagree that our contributions are not novel. We rely on the well-established leave-one-out principle of measuring memorization because it is notably the most rigorous definition of memorization that has been established (see references [6, 21,59]). Yet our contributions go well beyond simply applying the framework to graphs.
>
> Specifically:
> 1. To the best of our knowledge and as acknowledged by Reviewer FwDg, our paper is the first work to comprehensively analyze memorization in semi-supervised node classification;
> 2. We show a novel inverse relationship between memorization and graph homophily and link this to the GNNs’ implicit bias in leveraging graph structure during training;
> 3. We proposed a new metric, i.e., LDS to show that nodes with higher label inconsistency in their feature-space neighborhood are significantly more prone to memorization;
> 4. We empirically analyze the link between GNN memorization and privacy risks, highlighting the real-world impact of memorization on GNNs;
> 5. We also propose a graph rewiring framework as an initial strategy to mitigate memorization. The evaluation results show that it can reduce memorization while maintaining model utility. Further incorporating comments from Reviewers FwDg and uuwn, we have also added additional experiments:
>
> i. Extended our framework to the graph classification setting on 4 datasets and showed promising connections to node feature noise and memorization rate.
>
> ii. Additional experiments on Graph Transformers and large datasets such as ogbn-arxiv.
>
> In summary, we provide an **end-to-end analysis of the phenomenon of memorization in GNNs**. Our work not only shows that GNNs memorize node labels, but also explains its emergence and provides an initial strategy via graph rewiring to mitigate memorization. Finally, we are also the first to show its practical implications for privacy risks.
>
> >**Is it appropriate to define the GNN's memorization based on a single node (i.e., leave-one-out)?**
>
> It is appropriate to define the memorization in GNNs based on a single node, considering that the task is a *node classification task*. We then analyze the relationship between memorization and complex graph-specific elements, including graph-level and node-level characteristics and properties, e.g., graph homophily and LDS. In the response to Reviewer FwDg, we additionally show that our framework is not limited to instantiation on single nodes: To measure memorization for graph-level tasks, we show that our framework seamlessly extends to leave-one-out style definitions over entire graphs.
>
> >**I suggest adding research about the 2-hop topology on graphs**
>
> We interpret the reviewer’s suggestion as a valuable question about whether our findings on memorization are robust to the GNN's receptive field size, or if they are merely an artifact of a 1-hop message passing scheme. To investigate this, we conducted additional experiments with another GNN variant called Simplified Graph Convolution (SGC) [A], which is designed to analyze the impact of multi-hop neighborhoods by propagating features over a specified number of hops, K. We ran our experiments using SGC with K=2, thereby allowing the model to aggregate information directly from the 2-hop neighborhood and present results on 4 datasets in the following table.
>
> `SGConv:`
> |Datasets|AvgMemScore|MR(%)|
> |---|---|---|
> |Cora|0.0482±0.0165|3.8±0.4|
> |Citeseer|0.1124±0.0213|9.7±2.8|
> |Chameleon|0.3852±0.0111|37.3±0.8|
> |Squirrel|0.3745±0.0141|34.5±3.2|
>
> The results conclusively show that our core findings are robust to this change: Concretely, we observe the same strong inverse relationship between graph homophily and memorization rate when using a 2-hop aggregator. Low-homophily graphs (i.e., Chameleon and Squirrel) continue to exhibit significantly higher memorization, confirming that our conclusions are not dependent on a limited receptive field but are a more fundamental property of how GNNs learn on these graphs. We added these SGC results and a discussion to Appendix E to demonstrate the robustness of our framework.
>
> >**Would it be better to fit the results reflected in Fig. 5 and 7 using a quadratic polynomial?**
>
> We thank the reviewer for the suggestion to explore polynomial fitting for the trends in Figures 5 and 7. Our objective in presenting these figures is to highlight the inverse relationship that exists between the alignment kernels and the memorization rate. Additionally, we believe that fitting a specific functional form would risk overfitting to the data and potentially introduce spurious artifacts. Our Pearson correlation analysis from the paper, which makes no assumptions about the parametric form of the relationship, already shows correlation coefficients r=-0.98 and r=-0.88 for synthetic datasets (presented in Section 4.2, Proposition 3) and r=-0.79 and r=-0.58 for real-world datasets (presented in Section 5.1) respectively strongly reinforcing our claims.
>
>
> **References:**
>
> [A] Simplifying Graph Convolutional Networks. Wu et al. ICML 2019.

---

> > ### Comment · Reviewer_UeH1 · 2025-08-05
> >
> > Thank you for your detailed response.  I think my main concerns are not resolved by the authors. Firstly, I think the memorization score defined in Eq. (1) is not appropriate for graph-structured data, especially for homophilous graphs, since nodes are connected (non-IID) and tend to have the same label. For homophilous graphs, even without seeing node $v_i$ in the training data, GNNs can correctly predict its label if seeing the neighboring nodes of  $v_i$. Nevertheless, for heterophilic graphs, since connected nodes tend to have different labels, Eq. (1) may be useful. Thus, Proposition 1 is ordinary. Secondly, Label Disagreement Score (LDS) is very similar to the Homophily ratio defined in Geom-GCN. Thirdly, Figures 3 and 5 show linear relationships. Could they be confounders? Can the authors show causal relationships? In addition, the GNNs and the datasets used in the paper are very ordinary. Why didn't the authors study heterophilic GNNs such as H2GCN and Geom-GCN? Finally, using the existing method of graph rewiring proposed in [51] as a means to mitigate memorization is not innovative, either.

---

> > > ### Author Response · Authors · 2025-08-05
> > > **Further clarifications on the rebuttal**
> > >
> > > We thank the Reviewer for engaging in the rebuttal and are happy to further clarify.
> > >
> > > >**Equation 1: For homophilous graphs, even without seeing node $v_i$ in the training data, GNNs can correctly predict its label if seeing the neighboring nodes of $v_i$. Nevertheless, for heterophilic graphs, since connected nodes tend to have different labels, Eq. (1) may be useful.**
> > >
> > > We have demonstrated that **our memorization definition is applicable to homophilic graphs**. In fact, our metric captures precisely the point raised by the Reviewer.
> > >
> > > **High-homophily:** When a GNN can *generalize from neighbors* (as in high-homophily graphs), it **does not need to memorize**. Our framework correctly captures this by yielding a **low memorization score**. Since real-world graphs, such as Cora, are not perfectly homophilic, our results reflect this with a low but non-zero memorization rate.
> > >
> > > **Low-homophily:** When a GNN *cannot generalize* from neighbors (as in low-homophily/heterophilic graphs), it is forced to **memorize** the node's label to minimize training loss. Our framework correctly captures also this case by yielding a **high memorization score** for graphs like syn-cora-h0.0 and Chameleon.
> > >
> > > Prior to our work, this inverse relationship between homophily and memorization was neither established nor measured. Our contribution lies in formalizing this concept, providing a framework to measure it, and explaining the mechanisms behind it.
> > >
> > > >**Label Disagreement Score (LDS) is very similar to the Homophily ratio defined in Geom-GCN**
> > >
> > > Our Label Disagreement Score is fundamentally different from the homophily ratio ($\beta$) proposed in Geom-GCN. The homophily ratio defines the fraction of edges connecting nodes of the same class in a graph, which is a global measure for the **entire graph**; this fails to provide any insights into why only certain nodes in a graph are highly susceptible to memorization.
> > >
> > > To address this gap specifically, we propose our Label Disagreement Score (LDS) which mainly finds the **local-anomaly** in the label-feature space of the graph. Specifically, our metric uses the $l_2$ distance between the node features to construct a k-NN graph where $k$ is set to 3 in our experiments to analyse the local *3-hop neighbourhood around a candidate node* that has exhibited memorization to check how many of its neighbour’s labels agree/disagree, effectively identifying atypical nodes that are highly susceptible to memorization.
> > >
> > > In summary, the homophily ratio measures a **full-graph property**, our LDS measures a **local property**.
> > >
> > > >**Figures 3 and 5 show linear relationships: confounders / causal relationships?**
> > >
> > > We thank the reviewer for raising this point. Figures 3 and 5 are meant to illustrate consistent empirical trends, not to imply causality. While causal analysis can be valuable, it requires additional assumptions or interventional data beyond the scope of this work. We have clarified in the text that these are observational findings and not causal claims.
> > >
> > > >**Heterophilic GNNs such as H2GCN and Geom-GCN**
> > >
> > > In the paper, we showed extensive experiments on **3 commonly used GNN backbones GCN, GraphSAGE and GATv2 on 9 real-world datasets + 5 synthetic datasets**. Additionally as part of the rebuttal we added results for **4 new large datasets, results on Graph Transformer and SGC GNN model, and graph classification on 4 datasets**. To address the reviewer’s additional comment, we ran more experiments and measured memorization on H2GCN-2hop:
> > >
> > > `H2GCN-2hop`
> > >
> > > |**Datasets**|**AvgMemScore**|**MR(\%)**|
> > > |-|-|-|
> > > |Cora|0.1547±0.0193|15.1±1.0|
> > > |Citeseer|0.2641±0.0093|26.5±0.5|
> > > |Chameleon|0.4793±0.0296|47.8±1.1|
> > > |Squirrel|0.6441±0.0102|65.6±1.9|
> > >
> > > Our proposed framework is still applicable and shows the same trends, namely that the homophilic graphs (Cora, Citeseer) experience lower memorization than the heterophilic ones (Chameleon, Squirrel). We are currently running experiments on Geom-GCN, and will include the results in the final paper.
> > >
> > > >**Innovativeness of graph rewiring**
> > >
> > > We again would like to highlight that our work is the first to systematically study memorization in GNNs, and that, to the best of our knowledge, no work before attempted to mitigate memorization. Our innovation in terms of mitigation lies in identifying a problem (memorization in GNNs), understanding the problem from first principles (the relation with higher homophily leading to lower memorization), and finally, based on the fundamental understanding of the causes, identifying the suitable solution, (increase homophily). We find that increasing homophily **is effectively achieved through rewiring**. In the answer to Reviewer RAPS, we evaluate this method against other baselines and show that it works best, i.e., maintains utility while reducing memorization. Hence, while rewiring itself is not new, **its use as a memorization mitigation strategy rooted in our homophily analysis is completely novel**.

---

> > > > ### Comment · Reviewer_UeH1 · 2025-08-05
> > > >
> > > > Thank you very much for your clarifications. Let me further clarify my concerns:
> > > >
> > > > 1. Due to the non-IID properties and structural dependencies in graphs, the feature and label of each node depend not only on the node itself but also on its neighbors. Therefore, simply removing a node cannot fully isolate the node's influence on the model. Thus, I think Eq. (1) is not appropriate for homophilous graphs. I think there are more aspects worth exploration, for example:
> > > >
> > > > (1) Neighbor Information Shielding: When performing the leave-one-out, not only should the node’s label be removed, but the labels of its neighbors (or at least some of them) should also be shielded to prevent information leakage.
> > > >
> > > > (2) Structural Perturbation: While removing node label, the neighborhood structure should be randomly perturbed (e.g., by randomly deleting or adding edges) to assess whether the model relies on a specific structure for memory.
> > > >
> > > > 2. LDS is actually a local homophily ratio for a node.
> > > >
> > > > 3. I think linear or inverse relationship is some kind of superficial phenomenon, which means the graph structure (homophily) may be the confounder of the memorization rate. Thus, I think there needs more explorations.

---

> > > > > ### Author Response · Authors · 2025-08-06
> > > > > **Further Clarifications**
> > > > >
> > > > > We thank the Reviewer for further engaging in the rebuttal and for bringing up these questions. We are happy to address them one-by-one below
> > > > >
> > > > > >**Due to the non-IID properties and structural dependencies in graphs, the feature and label of each node depend not only on the node itself but also on its neighbors. Therefore, simply removing a node cannot fully isolate the node's influence on the model. Thus, I think Eq. (1) is not appropriate for homophilous graphs.**
> > > > >
> > > > > To address this question, we would like to invite the Reviewer to consider the *standard setup of semi-supervised node classification*, which is based on **a single graph containing train, validation, and test nodes simultaneously**. In this setting, label information is restricted via *masking*: the model is trained only using the labels of training nodes and evaluated solely on the labels of test nodes. However, the full graph structure, including edges between training and test nodes, is accessible to the model during training.
> > > > >
> > > > > By the logic of the Reviewer’s argument, the mere existence of such edges would imply a form of train/test leakage. This would put into question the validity of the entire semi-supervised node classification learning, as currently practiced in the literature.
> > > > >
> > > > > Our framework operates on the same principle. For model $f$, we include the label of node $v_i$. For model $g$, we use a mask to exclude the label of the node $v_i$. In both cases, the underlying graph structure is held constant, because the goal is to measure the effect of including one additional labeled sample in the training process, while holding all other factors (including the graph topology) equal. Under this setup, the prediction of model $g$, trained without $v_i$’s label, on $v_i$ represents the model’s pure generalization, namely its ability to predict $v_i$ based solely on its features and its structural context/neighbouring nodes within the graph. The prediction of model $f$, trained with $v_i$’s label, in contrast, represents a combination of that same generalization power *plus* **any memorization of $v_i$’s specific label $y_i$**. Therefore, our definition in Eq. (1) is not just appropriate; we believe it is the only definition that correctly aligns with the standard assumptions of semi-supervised learning on graphs.
> > > > >
> > > > > >**I think there are more aspects worth exploration [...]:
> > > > > (1) Neighbor Information Shielding: When performing the leave-one-out, not only should the node’s label be removed, but the labels of its neighbors [...]. (2) Structural Perturbation: While removing node label, the neighborhood structure should be randomly perturbed (e.g., by randomly deleting or adding edges) [...]**
> > > > >
> > > > > These are interesting future directions. However, applying shielding or structural perturbations would significantly alter the original graph, potentially introducing information loss and unintended artifacts. This, in turn, could confound the interpretation of the memorization signal, as it becomes unclear whether observed effects stem from the removal of the target node’s information or from the broader disruption of the graph’s structure. Careful disentanglement of these factors would be required to make sure that such modifications yield meaningful insights into memorization.
> > > > >
> > > > > >**LDS is actually a local homophily ratio for a node**
> > > > >
> > > > > We are happy to elaborate on the difference between a naive local homophily ratio and our LDS: Traditional homophily ratios operate on the graph's predefined topology using only labels, asking "Do my connected neighbors have my label?" Our LDS operates in feature space using both features and labels, asking "Do nodes with similar features to mine also have my label?" This distinction is crucial because GNNs operate on both graph structure and node features. A node can be anomalous (requiring memorization) when its features are inconsistent with labels, even if its neighbors are homophilic. Our LDS detects this inconsistency, which topological homophily measures miss. Therefore, LDS is not just a local version of homophily, it's a feature-aware metric for characterizing nodes susceptible to memorization.
> > > > >
> > > > > >**I think linear or inverse relationship is some kind of superficial phenomenon, which means the graph structure (homophily) may be the confounder of the memorization rate. Thus, I think there needs to be more explorations.**
> > > > >
> > > > > We have validated that the inverse relationship holds over various diverse setups, namely 6 distinct GNN architectures: GCN, GraphSAGE, GATv2, SGC, H2GCN, and a Graph Transformer, 13 real-world datasets (9 in the paper, 4 in the rebuttal) and 5 synthetic datasets with controlled homophily levels. The fact that **this trend holds universally across such a wide variety of models and data conditions** represents, to our belief, a thorough exploration. However, if the Reviewer has other concrete suggestions and experiments they would like to see explored, we are happy to run them and provide the results.

---

### Official Review · Reviewer_uuwn · 2025-07-02

**Clarity:** 3
**Significance:** 2
**Originality:** 3
**Rating:** 4
**Confidence:** 3

**Summary:**

This paper proposes NCMemo, a framework to measure how much GNNs memorize training labels in node classification tasks. This paper finds that memorization increases as graph homophily decreases. The authors show that this happens because GNNs rely more on memorizing labels when the graph structure provides less useful information. They also propose a Label Disagreement Score (LDS) to identify which nodes are more likely to be memorized. Finally, they show that graph rewiring can reduce memorization and its associated privacy risks without hurting model performance.

**Questions:**

Please see weaknesses

**Ethical Concerns:**

["NO or VERY MINOR ethics concerns only"]

**Final Justification:**

My major concerns are addressed by the authors. The technical part directly follows the leave-one-out memorization framework, which is standard. While, I appreciate the novel insights of GNN memorization from their empirical studies. Hence, I tend to accept the paper.

**Limitations:**

yes

**Quality:**

3

**Strengths And Weaknesses:**

Strengths:
- This paper focuses on memorization in GNNs. This helps fill an important gap in understanding generalization and privacy risks in graph learning.
- This paper proposes a new framework to quantify per-node memorization, inspired by the leave-one-out memorization framework.
- This paper provides insightful results on the relationship between homophily, memorization, and label inconsistency, based on both synthetic and real-world datasets.
- This paper includes graph rewiring to mitigate memorization and privacy risks.

Weaknesses:
- This paper highlights the difficulty in conceptualizing and defining per-sample memorization, but the proposed NCMemo framework seems not including graph-specific techniques to tackle the interactions of graph elements.
- The framework relies on training paired models with/without each node (or partition approximations), which may be computationally expensive. The scalability of NCMemo on large-scale graphs in the real world can be limited.
- More datasets and advanced GNN backbones can be included in the experiments such as open graph benchmarks and graph transformers.

---

> ### Author Rebuttal · Authors · 2025-07-30
>
> We would like to thank the Reviewer for the detailed feedback, and for acknowledging our work’s contribution in filling an *important gap in the understanding of generalization and privacy in graph learning*, and for providing *insightful* results on the relationship between homophily, memorization, and label inconsistency. Below, we address the comments one by one.
>
> >**The proposed NCMemo framework seems not including graph-specific techniques to tackle the interactions of graph elements.**
>
> Our framework builds on Feldman's leave-one-out definition [21], which is the *de facto* standard for studying memorization in classification tasks. Since our focus is *node classification*, this choice provides a well-established and principled foundation. By relying on our framework, we obtain multiple *graph-specific* findings:
>
> i. We propose a novel inverse relationship between the node label memorization and graph homophily, the property of graph which directly takes into account if two connected nodes have the same labels.
>
> ii. Our second novel contribution explains the emergence of memorization in GNNs through the lens of Neural Tangent Kernel and alignment matrices such as Kernel-Graph and Kernel-Target alignment [61] all of which show the implicit bias of GNNs to use the graph structure during the training.
>
> iii. Our novel metric, the Label Disagreement Score, characterizes node atypicality by analysing the local anomaly in the node label. The node feature space allows us to precisely point out which nodes are highly susceptible to memorization.
>
> iv. Our initial strategy to mitigate memorization leverages graph rewiring, that directly involves manipulating the edges of the graph based on feature similarity.
>
> All of these elements *directly use graph-specific elements and provide novel insights into graph-specific memorization*.
>
> >**Necessity of model pairs for assessing memorization and overhead**
>
> The leave-one-out memorization is theoretically well-founded and provides a precise notion of per-sample-level memorization. While the ideal form requires training multiple models, in our work, we **approximate it efficiently**: Our method only requires **a single additional trained model** and evaluates the effect for simultaneous removal of batches of nodes (25% of training nodes). This keeps the overhead minimal while still aligning closely with the theoretical definition. We also presented the empirical runtime for training models $f$ and $g$, calculating the memorization scores for the candidate set and calculating the LDS in **Tables 14-16** in Appendix H.3. For instance, on a larger graph such as Squirrel it takes 6.23±0.21 seconds to obtain memorization scores and 24.49±0.11 seconds to calculate the LDS.
>
> >**More datasets and GNN backbones, e.g., open graph benchmarks and graph transformers**
>
> We have added experiments on larger graph datasets, including ogbn-arxiv. The results are presented in the table below. We observe that the average memorization score and memorization rate on these datasets are low because the graphs are highly homophilic, which aligns with our Proposition 1, i.e., memorization is low for highly homophilic graphs. We also present the empirical runtimes in seconds for training models f and g and obtaining the memorization scores for the candidate set.
>
> `Large Datasets:`
>
> |Datasets|AvgMemScore|MemRate(%)|Nodes|Edges|Homophily|Runtime(seconds)|
> |---|---|---|---|---|---|---|
> |Photo|0.0093±0.0084|0.1±0.0|7,650|238,162|0.7850|3.97±0.24|
> |Physics|0.0156±0.0018|0.4±0.1|34,493|495,924|0.8724|17.82±0.24|
> |Computer|0.0005±0.0234|0.6±0.9|13,752|491,722|0.6823|6.26±0.19|
> |ogb-arxiv|0.0031±0.0051|0.4±0.3|169,343|1,166,243|>0.5|491.01±1.55|
>
> We also run additional experiments on a more advanced GNN backbone, a **Graph Transformer**, on four real-world datasets. We use the graph transformer setup proposed in [A] with slight modifications, namely we set the attention heads to 2 (instead of 8) and use a 1 layer graph transformer (Input → Linear Projection → [Attention + FFN Block] → Output Linear). The results are presented in the table below, along with the runtimes in seconds for training models f and g and obtaining the memorization scores for the candidate set. The results show that our memorization framework can be applied seamlessly to other GNN architectures including powerful architectures such as Graph Transformers, and that the propositions proposed in the main paper still hold.
>
> `Graph Transformer:`
>
> |Datasets|AvgMemScore|MR(%)|Runtime|
> |---|---|---|---|
> |Cora|0.1359±0.0072|13.4±0.7|1355.43±3.76|
> |Citeseer|0.2486±0.0070|24.9±0.20|1158.74±7.11|
> |Chameleon|0.2807±0.0150|27.3±1.3|1222.66±1.49|
> |Squirrel|0.4810±0.0094|47.7±1.2|4748.68±41.52|
>
>
> **References:**
>
> [A] A critical look at the evaluation of GNNs under heterophily: Are we really making progress? Platonov et al. ICLR 2023.

---

> ### Author Response · Authors · 2025-08-05
> **Request to respond to the rebuttal**
>
> We would like to thank the Reviewer again for their constructive comments. Based on their suggestions, we have additionally:
>
> (1) clarified the novelty and contributions of our work – proposing a framework built on Feldman’s leave-one-out definition, and relying on our framework, we obtained multiple *graph-specific* findings;
>
> (2) explained our evaluation strategy, which follows [59] to keep the overhead minimal while still aligning closely with the theoretical definition;
>
> (3) run experiments on larger graph datasets, including *ogbn-arxiv*, and a more advanced GNN backbone, i.e., *Graph Transformer*, confirming that our memorization framework can be applied seamlessly to larger graph datasets and other GNN architectures, and the propositions still hold. We also included empirical runtime analysis to demonstrate the scalability of our approach.
>
> These additions have significantly strengthened our contribution. We believe these enhancements have substantially improved the submission and would be happy to clarify any remaining questions the Reviewer may have. We kindly ask them to reassess their rating in light of these improvements.

---

> > ### Comment · Reviewer_uuwn · 2025-08-05
> >
> > I thank the authors for providing further clarifications and experimental results. Although this paper delivers several novel graph-specific insights on memorization, the techniques used to study memorization seem to be a direct adaptation of the leave-one-out memorization to graphs. Hence, I still think the technical challenges of memorization on graphs are limited, which slightly undermines the contributions and significance of the work. Meanwhile, I appreciate the new insights on the connections between homophily and node memorization. As the complementary results address my concerns on scalability, I am willing to increase my score to positive.

---

> > > ### Author Response · Authors · 2025-08-06
> > > **Thanks for acknowledging the rebuttal**
> > >
> > > We thank the Reviewer for engaging in the discussion and for the recognition of our additional results. Below we would like to clarify some of the details
> > >
> > > >**the techniques used to study memorization seem to be a direct adaptation of the leave-one-out memorization to graphs**
> > >
> > > As stated in our initial response, Feldman's leave-one-out definition is the *gold standard* for studying memorization in classification tasks. Given that we operate on *node classification*, relying on this framework represents the most principled foundation that the memorization community has agreed on. Using our adaptation to graphs, we provided multiple *graph-specific* findings and are happy that the Reviewer appreciates our “new insights on the connections between homophily and node memorization”.
> > >
> > > >**I still think the technical challenges of memorization on graphs are limited**
> > >
> > > While adapting the leave-one-out framework to graphs may not, in itself, pose a major technical challenge, we believe the strength of our work lies in the novel insights it makes possible. Specifically, we contribute (1) a principled framework to quantify memorization in GNNs, (2) a thorough evaluation on how homophily affects memorization, (3) a connection to the resulting privacy risks, and (4) a mitigation strategy grounded in structural understanding.
> > >
> > > > **Meanwhile, I appreciate the new insights on the connections between homophily and node memorization.**
> > >
> > > We are happy to see that these new insights are appreciated.
> > >
> > > >**As the complementary results address my concerns on scalability, I am willing to increase my score to positive.**
> > >
> > > As the Reviewer mentioned that they are now inclined to increase their score to positive, we would be very grateful *if they could kindly update their score on OpenReview to reflect this assessment*.

---

### Official Review · Reviewer_FwDg · 2025-07-03

**Clarity:** 4
**Significance:** 4
**Originality:** 4
**Rating:** 6
**Confidence:** 3

**Summary:**

The authors present a framework for quantifying label memorization in graph neural networks trained on semi-supervised node-classification tasks. They show that GNNs depend on memorization when operating on graphs with low homophily (i.e. graphs whose connected nodes do not share similar features). They show that high memorization leads to poor generalization via a kernel-target alignment metric. Finally, they reveal that the reason for increased memorization/poor performance for low homophily graphs is due to the tendency of GNN models to use structural information so in low homophily regimes,the lack of information from graph structure leads to more label memorization.

**Questions:**

### Questions
- Is it possible to show experimentally how memorization leads to poor generalization (as suggested by Prop. 2)?
- NCMemo is specifically for semi-supervised node classification but can the authors say a few words on if this framework can also be extended to graph classification tasks? In that case, is it possible to analyze the intermediary embeddings from each GNN layer to draw conclusions regarding how GNNs memorize information for classification tasks?

**Ethical Concerns:**

["NO or VERY MINOR ethics concerns only"]

**Final Justification:**

I repeat the comment that I made here:

I have read the rebuttal to my comment as well as the other reviews and rebuttals. They have addressed all of my questions. This paper is the first to systemically study memorization in graph neural networks and the authors have presented strong empirical evidence both for when GNNs will memorize information (low homophily regimes) as well as how memorization will affect the performance on the GNN. While their NCMemo framework probably could be improved further, I think the aforementioned findings in the current manuscript will already significant enough to for me to strongly recommend acceptance to NeurIPS. The framework itself is also general enough to be applied to many different graph learning techniques and I think it is fair to mainly consider standard message-passing GNNs (GCN, GAT, GraphSAGE) as they are still widely used in practice. I agree with other reviewers that graph transformers have become much more popular lately and I appreciate the additional experiments that the authors have carried out for graph transformers. Additionally, I do not think the authors should be faulted for lack of theory or proof. I think it is equally as significant to do many careful and thoughtful experiments to examine some phenomena (which has not been previously explored) and use such empirical investigations to understand when/where to use GNNs and how they might fail in harmful ways (their final section on privacy risks).

All this to say: I maintain my score.

**Limitations:**

yes

**Quality:**

4

**Strengths And Weaknesses:**

### Strengths

(1) *Significance*: This seems to be the first work on memorization in graph neural networks. Provides a good framework/metric for understanding how and when graph neural networks memorize node features and has potential implications for understanding GNN performance out of distribution (more memorization possibly leads to bad OOd generalization).
(2) *Extensive evaluation*: First, the authors do extensive experiments on both synthetic and real-world datasets. These experiments help clearly explain the link between graph homophily and memorization. They show a clear relationship between homophily and their metric which measures the rate of memorization for GNNs. In their appendix, they also include extensive evaluation with GAT and GraphSAGE (two commonly used graph architectures). I also appreciated the section analyzing privacy risks and the real-world impact of memorization.

In general, this is a strong paper which I think will have significant impact especially since it seems to be the first work on memorization in GNNs. The authors provide a convincing argument for studying GNN memorization from studying theoretically *when* we see memorization occurs to experimentally demonstrating the problems that arise from meorization at NN deployment time.

### Weaknesses

(1) Their framework only works in the node-classification regime (which is still important). I wonder if the authors could say a few words on how we should keep track of/understand memorization when considering graph level classification tasks.
(2) Proposition 2 suggests that GNNs suffer from poor generalization when there is high memorization. This part seems relevant esp. given the increased attention to out-of-distribution generalization in several previous works so I wonder if the authors can elaborate more on this point and show this property experimentally?

---

> ### Author Rebuttal · Authors · 2025-07-30
>
> We thank the Reviewer for recognizing the significance of this work as the first systematic study of memorization in graph neural networks. We appreciate the acknowledgment of our extensive experimental evaluation and the clear relationship established between graph homophily and memorization. Below we address the Reviewer’s comments and questions.
>
> >**The framework targets the node-classification regime. How about graph-level classification?**
>
> We conducted additional analyses to apply our NCMemo framework to graph classification tasks. Concretely, we change the dataset S to contain graphs rather than nodes, and measure memorization as the difference on expected behavior between models $f$ trained on $S$ and models $g$ trained on $S \setminus gr_i$, where $gr_i$ is the graph whose memorization we want to measure.
>
> We applied this framework to measure memorization on 4 graph classification datasets, namely the MUTAG, AIDS, BZR and COX2 [A], using a GCN backbone.  The data split strategy is similar to the one presented in the paper, where we divide the graph dataset into 60%/20%/20% as training/testing/validation sets, respectively. Among the 60% training set, we further randomly divide the datasets into three disjoint subsets, i.e., shared graphs $S_S$, candidate graphs $S_C$, and independent graphs $S_I$, with ratios 50%, 25% and 25%, respectively. We measure the average memorization scores on the candidate graphs, we obtained the following results:
>
>
> |Dataset|AvgMemScore|MR(%)|
> |---|---|---|
> |MUTAG|0.0011±0.0045|0|
> |AIDS|0.0767±0.0260|0|
> |BZR|-0.0096±0.0067|0|
> |COX2|0.0068±0.0241|0|
>
> These initial results suggest that there is no memorization happening in these training setups. We hypothesise that this is because there are no *outlier* graphs, making it possible to learn without memorization, following the reasoning of Feldman [21].
>
> To test this hypothesis and assess whether our framework is sensitive to measure memorization in graph tasks when it does appear, we conducted additional experiments where we introduce Gaussian noise to the node features of all the nodes of candidate graphs ($S_C$), de facto turning them into outliers, and see whether their experienced memorization increases. Our results in the tables below support this hypothesis and show that the more *outlier* the graphs become through the addition of more noise, the more memorization they experience.
>
> `GCN+MUTAG:`
>
> |NoiseLevel|AvgMemScore|MR(%)|
> |---|---|---|
> |0.0|0.0011±0.0045|0|
> |0.1|0.0862±0.0244|2.2±1.9|
> |0.3|0.2352±0.0744|18.3±9.9|
> |0.5|0.3419±0.0420|33.3±3.7|
> |0.7|0.3680±0.0239|39.8±4.9|
> |0.9|0.4544±0.0302|46.2±6.7|
>
> `GCN+AIDS:`
>
> |NoiseLevel|AvgMemScore|MR(%)|
> |---|---|---|
> |0.0|0.0066±0.0285|0|
> |0.1|0.0767±0.0260|0|
> |0.3|0.1351±0.0298|6.5±3.2|
> |0.5|0.2086±0.0295|13.7±1.2|
> |0.7|0.2465±0.0087|18.8±1.4|
> |0.9|0.2367±0.0346|17.6±2.2|
>
> `GCN+BZR:`
>
> |NoiseLevel|AvgMemScore|MR(%)|
> |---|---|---|
> |0.0|-0.0096±0.0067|0|
> |0.1|0.1993±0.0137|9.2±3.1|
> |0.3|0.2965±0.0064|27.7±0.0|
> |0.5|0.2091±0.0109|15.9±0.9|
> |0.7|0.1790±0.0010|18.5±0.0|
> |0.9|0.1998±0.0006|20.0±0.0|
>
> `GCN+COX2:`
>
> |NoiseLevel|AvgMemScore|MR(%)|
> |---|---|---|
> |0.0|0.0068±0.0241|0|
> |0.1|0.1939±0.0116|2.6±1.3|
> |0.3|0.2326±0.0145|21.5±1.5|
> |0.5|0.2856±0.0012|28.9±0.0|
> |0.7|0.2237±0.0101|23.7±0.0|
> |0.9|0.1838±0.0092|18.4±0.0|
>
> While more future work will be required to understand the mechanisms that cause graph-level memorization, we believe that our framework’s flexibility to support this kind of analysis, makes it a valuable tool for future work in this direction.
>
>  >**Showing Proposition 2 experimentally (memorization leads to poor generalization)**
>
> Our paper provides direct experimental evidence for Proposition 2, showing that higher memorization is linked to poorer generalization. This is demonstrated through two key metrics:
>
> 1. In Figures 4c and 8c of the paper we present the Kernel-Target alignment (Yang et al [61]) which is a formal measure of generalization. This alignment is poor for heterophilic graphs (which have high memorization) and the alignment is high for homophilic graphs directly providing evidence for *high memorization results in poor generalization.*
>
> 2. Further, we also present the test accuracies for real-world datasets in the table below (in addition to the accuracies reported for synthetic datasets in Appendix G), which clearly show heterophilic datasets, such as Chameleon and Squirrel, have lower generalization performance compared to homophilic datasets such as Cora and Citeseer while having higher memorization rates.
>
> `Generalization Performance of Real-world Datasets:`
>
> |Datasets|Test Accuracy of Model f|AvgMemScore|MR(%)|
> |---|---|---|---|
> |Cora|85.24±0.51|0.1033±0.0006|9.7±0.1|
> |Citeseer|77.52±1.63|0.2661±0.0048|28.4±0.8|
> |Chameleon|33.92±0.80|0.5285±0.0053|52.9±0.3|
> |Squirrel|26.79±0.76|0.5050±0.0133|53.6±1.3|
>
>
>
>
> **References:**
>
> [A] TUDataset: A collection of benchmark datasets for learning with graphs. Morris et al, GRL+ Workshop, ICML 2020.

---

> > ### Comment · Reviewer_FwDg · 2025-08-01
> > **Response to authors**
> >
> > I have read the rebuttal to my comment as well as the other reviews and rebuttals. They have addressed all of my questions. This paper is the first to systemically study memorization in graph neural networks and the authors have presented strong empirical evidence both for *when* GNNs will memorize information (low homophily regimes) as well as *how* memorization will affect the performance on the GNN. While their NCMemo framework probably could be improved further, I think the aforementioned findings in the current manuscript will already significant enough to for me to strongly recommend acceptance to NeurIPS. The framework itself is also general enough to be applied to many different graph learning techniques and I think it is fair to mainly consider standard message-passing GNNs (GCN, GAT, GraphSAGE) as they are still widely used in practice. I agree with other reviewers that graph transformers have become much more popular lately and I appreciate the additional experiments that the authors have carried out for graph transformers. Additionally, I do not think the authors should be faulted for lack of theory or proof. I think it is equally as significant to do many careful and thoughtful experiments to examine some phenomena (which has not been previously explored) and use such empirical investigations to understand when/where to use GNNs and how they might fail in harmful ways (their final section on privacy risks).
> >
> > All this to say: I maintain my score.

---

> > > ### Author Response · Authors · 2025-08-02
> > > **Thank for acknowledging the rebuttal**
> > >
> > > We thank the reviewer for their engagement during the rebuttal phase and for carefully considering not only our responses to their own questions but also our replies to other reviewers. We greatly appreciate their positive feedback and remain available should any further questions arise.

---

### Author Response · Authors · 2025-08-07
**Rebuttal Summary**

As the rebuttal period is coming to an end, we would like to thank all the reviewers again for their time and effort in reviewing our work and providing detailed feedback. We thank Reviewer FwDg for noting that our work is a *”strong paper which I think will have significant impact especially since it seems to be the first work on memorization in GNNs”*, and for summarizing that *”the authors have presented strong empirical evidence both for when GNNs will memorize information (low homophily regimes) as well as how memorization will affect the performance on the GNN”*.

In general, Reviewers FwDg, uuwn, UeH1, and RAPS find our work novel and acknowledge its importance in filling an existing gap, e.g., Reviewer uuwn summarizes our work to *“fill an important gap in understanding generalization and privacy risks in graph learning”*. Finally, all reviewers appreciate our *”sufficient, and satisfactory”* (UeH1), *”insightful”* (uuwn), and *”extensive evaluation”* (FwDg), which leads to *“ robustness to the conclusions”* (RAPS).

We believe that the exchange with the reviewers during the rebuttal has helped us further improve our work significantly. We summarize the global highlights of our rebuttal here below:

1. **Extension of our Memorization Framework to Graph Classification**: We demonstrated that our memorization framework can be extended to graph classification and provided additional experiments on 4 graph datasets (MUTAG, AIDS, BZR and COX2).

2. **Scalability of our Framework**: We presented additional results on 4 new *large datasets*, namely Photos, Physics, Computer and ogb-arxiv to show how our framework can be scaled to large graphs. We have also included results on advanced architectures, namely Graph Transformer, H2GCN, and SGC. Our results on these architectures show that our core insights are model agnostic.

3. **Adding Baseline Mitigation Strategies**: In addition to graph rewiring, we also presented results on the utility of label smoothing and label smoothing+rewiring as alternative baselines to mitigate memorization. Our results show that rewiring is more effective in reducing memorization while also preserving model utility.

4. **Additional Insights on our Framework**: We provided additional detailed explanations of why the *leave-one-out strategy for memorization* works for our semi-supervised node classification setting, added a theoretical proof sketch of Proposition 2 and new experiments with different memorization threshold ($\tau$) values to demonstrate that our findings are agnostic to any specific threshold value.

We hope that these efforts reflect our strong commitment to incorporating the reviewers' suggestions and further improving our work.

Finally, we would like to express our gratitude to the AC for their time and effort in overseeing the review process of our submission.


With kind regards,

The Authors

---

### Decision · Program_Chairs · 2025-09-17

**Decision:**

Accept (poster)

**Comment:**

This paper presents a framework for quantifying label memorization in GNNs trained on semi-supervised node-classification tasks. The authors adopt the classic and standard leave-one-out memorization framework and give interesting insights on the relationship between homophily, memorization, and label inconsistency, based on both synthetic and real-world datasets.

One strength of the work is that it pioneers the study of memorization in GNNs, and it provides a metric for understanding how and when graph neural networks memorize node features.

The reviewers have a discussion on whether using the standard leave-one-out formula (Eq 1) is a good fit for graphs (particularly on node classification task). The issue arises because typically Eq 1 requires the i.i.d. assumption to quantify memorization, while the assumption does not hold in graphs in general.

Even though the definition of memorization (Eq 1) may not be perfect, it represents an initial step toward understanding memorization in message-passing GNNs and has the potential to inspire follow-up works in this new field. Therefore, I am inclined to recommend acceptance of this paper, provided that the authors include in the final version a clear discussion on both the rationale and the limitations of using Eq. (1) to quantify memorization in GNNs.